# Revealing nano-scale lattice distortions in implanted material with 3D Bragg ptychography

Peng Li [1,4], Nicholas W. Phillips [2,5], Steven Leake [3], Marc Allain [1], Felix Hofmann [2] & Virginie Chamard [1✉]

Small ion-irradiation-induced defects can dramatically alter material properties and speed up degradation. Unfortunately, most of the defects irradiation creates are below the visibility limit of state-of-the-art microscopy. As such, our understanding of their impact is largely based on simulations with major unknowns. Here we present an x-ray crystalline microscopy approach, able to image with high sensitivity, nano-scale 3D resolution and extended field of view, the lattice strains and tilts in crystalline materials. Using this enhanced Bragg ptychography tool, we study the damage helium-ion-irradiation produces in tungsten, revealing a series of crystalline details in the 3D sample. Our results lead to the conclusions that few-atom-large 'invisible' defects are likely isotropic in orientation and homogeneously distributed. A partially defect-denuded region is observed close to a grain boundary. These findings open up exciting perspectives for the modelling of irradiation damage and the detailed analysis of crystalline properties in complex materials.

[1] Aix-Marseille Univ, CNRS, Centrale Marseille, Institut Fresnel, Marseille, France. [2] Department of Engineering Science, University of Oxford, Parks Road, Oxford OX1 3PJ, UK. [3] ESRF – The European Synchrotron, 71 Avenue des Martyrs, 38000 Grenoble, France. [4] Present address: Diamond Light Source, Harwell Science and Innovation Campus, Fermi Ave, Didcot OX11 0DE, UK. [5] Present address: Paul Scherrer Institut, 5232 Villigen, Switzerland. ✉email: virginie.chamard@fresnel.fr

Atomic defects play a fundamental role in controlling the mechanical and physical properties of crystalline materials, resulting in critical hurdles for advanced applications, as epitomised in e.g. energy generation (nuclear[1] or photovoltaic[2]), energy storage[3], aerospace[4], micromechanics[5] and semiconductor miniaturisation[6]. Native or induced defects localise distortion of the crystal lattice and thereby reduce the overall strain energy[7]. Conversely, the behaviour of defects is strongly dependent on the microstructural environment, which provides fantastic potential for tuning material properties[8].

Understanding and exploiting defects in crystalline materials requires probing the material structure from atomic- to macroscale. At near atomic resolution, transmission electron microscopy (TEM) is an essential method, which allows the direct visualisation of lattice defects in two (2D) and even three dimensions (3D)[9,10]. While TEM is also sensitive to the associated lattice strains, their investigation is restricted to 2D and only possible for a small subset of samples (e.g. straight dislocations of edge character[11]). This is insufficient for understanding complex defect-defect interplays, since defects interact via their 3D strain fields and resulting stresses[12]. In inherently thin TEM samples, defects may also be lost to nearby surfaces that act as strong defect sinks, thus reducing the apparent defect density[13].

A further challenge concerns the visibility of small defects: In structures containing more than a few thousand atoms, TEM is insensitive to defects smaller than ~1.5 nm[14]. However crystals that contain a population of small defects with a broad size distribution constitute a large class of materials, such as those resulting from intense irradiation exposure (materials for future fusion and fission technologies[1], accelerator targets[15] or modified surfaces for biocompatibility[16]). Defects produced by irradiation range from single atom defects to clusters tens of nanometres in size, where the number density of defects is typically linked to the defect size by a power law with a negative exponent, i.e. the vast majority of defects are below the visibility limit[17]. Although small, these "invisible" defects can dramatically change the mechanical properties[18] or thermal transport behaviour[19], and may lead to irradiation-induced dimensional change[20,21]. Recent simulations have successfully predicted the presence and evolution of large populations of "invisible" defects during irradiation[22]. However, these predictions cannot be verified by existing experiments. As such there is an urgent need for techniques able to determine the "invisible" defect number density and spatial distribution.

A promising alternative is to infer the concentration of these small defects by measuring the distortions (i.e. strains) they cause in the crystal lattice. This idea has been successfully demonstrated using micro-beam Laue diffraction[20,21]. However the spatial resolution of this technique (~0.5 μm in 3D) is insufficient to resolve the nano-scale spatial heterogeneities and defect clustering predicted by simulations[22]. Although electron-diffraction based microscopy techniques could also indirectly measure these small defects via strain at very high spatial resolution, they are limited to 2D information and/or small fields of views.

In this context, we foresee that x-ray lens-less microscopy could play a major role[23]. Since its first demonstration in 2001[24], x-ray Bragg coherent diffraction imaging has been proven as a powerful method to investigate crystalline properties of materials[25] in various sample environments[26,27]. X-ray Bragg ptychography (BP), a recently developed method, combines sensitivity to atomic displacements of lattice planes and the outstanding imaging performances of ptychography. Thereby, it provides the means to image, in 3D, extended crystalline materials, with high sensitivity to weak crystalline displacements and strong robustness to large strain fields, those specifications being essential to cope with the distinctive crystalline features of defect-induced strains and native strains in e.g. irradiated materials.

To date, BP with 3D spatial resolution of 10–50 nm, and strain resolution on the order of $10^{-4}$ has been demonstrated[28,29], and applied to complex problems in material science, e.g. anti-phase domain boundaries in a metallic alloy[30], stacking faults and strain fields in semiconductor quantum wire[31,32] and crystalline domains in biominerals[33]. The advent of 4th generation synchrotrons, where BP will become available at several beamlines, should provide broader access to this microscopy method.

X-ray BP makes use of a series of 3D Bragg diffraction intensity datasets, measured in the vicinity of a Bragg reflection, obtained by scanning the sample across a localised illumination beam. The sample motion is designed to ensure a significant amount of probe overlap, the redundancy in the dataset allowing retrieval of the otherwise inaccessible phase of the diffracted field from the set of diffracted intensities[34]. As long as the probe is known, the phase recovery is performed with iterative algorithms, which further return the 3D sample image. An effective complex-valued crystalline electron density[35], $\rho(\mathbf{r})$, specifically designed to account for the Bragg geometry, is used to describe the sample as a function of $\mathbf{r}$, the 3D spatial coordinate. Its amplitude $|\rho(\mathbf{r})|$, provides information about the morphology (or density) of the scattering crystal domain. The spatially varying phase $\phi_{hkl}(\mathbf{r})$ associated with a specific hkl Bragg vector $\mathbf{Q}_{hkl}$ is linked to the atomic displacement field in the crystal, $\mathbf{u}(\mathbf{r})$, by $\phi_{hkl}(\mathbf{r}) = \mathbf{Q}_{hkl} \cdot \mathbf{u}(\mathbf{r})$.

However, current BP approaches suffer from substantial limitations. They require the acquisition of an extended dataset, which relies on the stability of the experimental set-up over long measurement times (typically 6–12 h). This requires samples to be resistant to radiation damage, a particular challenge for biologically relevant materials. Moreover, BP requires the 3D probe to be known prior to the sample crystalline electron density reconstruction, a strong limitation, which has not been overcome so far. Although some probe pre-characterisations can be performed via ptychography of a test pattern (or similar) in the transmission geometry, the final BP image quality strongly depends on the uncertainties introduced by the lack of detailed knowledge of the probe used during the BP experiment. In forward direction ptychographic imaging, this limitation has long been recognised, and simultaneous retrieval of the probe and object is now the universal measurement standard[36]. The additional complexity of the Bragg geometry has thus far prevented this refinement in BP.

Here, we use an advanced BP approach to investigate TEM-invisible defects in a tungsten-rhenium alloy sample implanted with helium ions. To enable these measurements, we develop a simultaneous probe refinement strategy that makes it possible to improve the retrieved image sensitivity and map a much larger field of view than previously attainable. Although single-crystalline samples present a constant electron density, the phase associated with lattice distortions in principle provides enough spatial diversity to separate the probe from the sample contribution. Notably, this was achieved without increasing the amount of collected data (and therefore the total acquisition time). This approach further allows us to extend the field view while improving the image quality, making it possible to directly compare implanted and non-implanted sample regions. This comparison highlights details of the crystalline structure, such as lattice damage from helium irradiation, several dislocations and sample preparation damage. Whilst we still cannot directly resolve 'invisible defects', we unambiguously probe them via the strain fields they cause, which extend over distances much larger than the defects themselves, thus capturing their presence and characteristics. As such, we can assess their behaviour in an unmatched manner. The results are discussed in the context of understanding irradiation damage processes within tungsten and

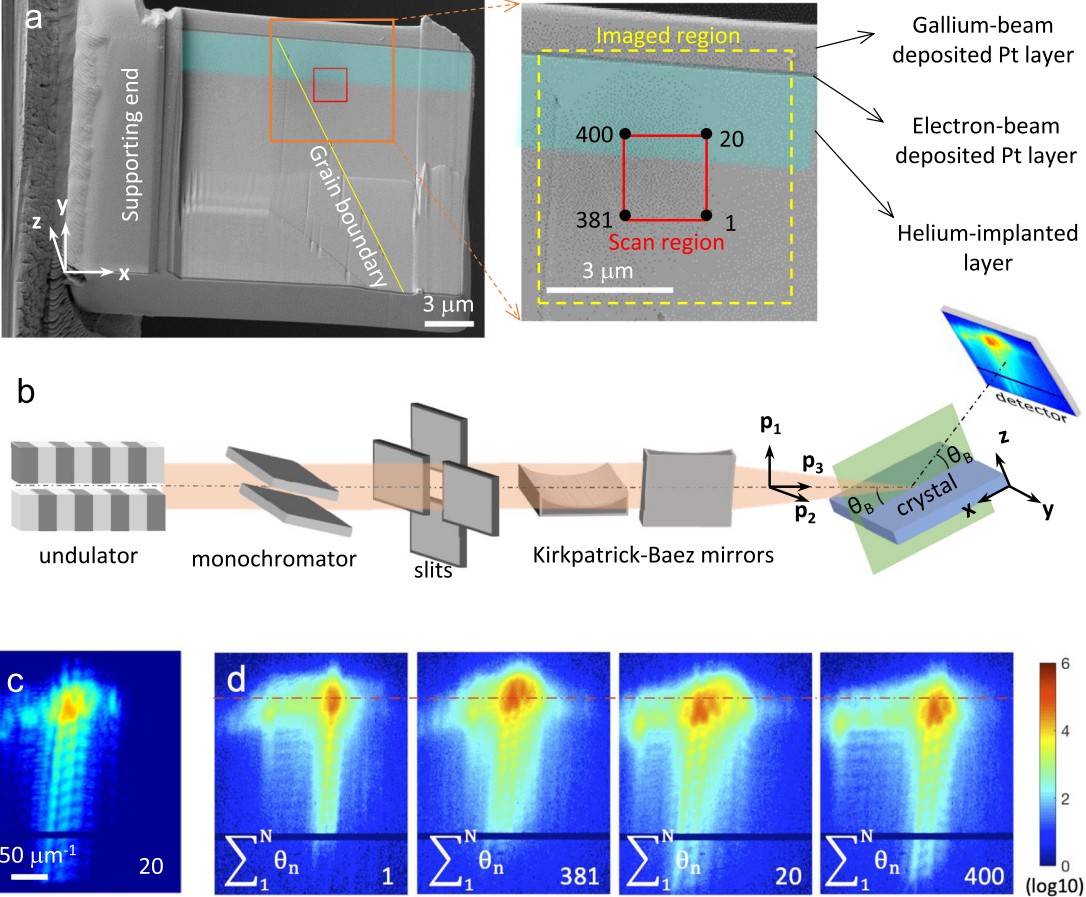

**Fig. 1 X-ray diffraction microscopy of the He-implanted tungsten foil. a** Scanning electron microscopy images of the tungsten foil sample. The platinum protection layers, helium-implanted layer and the grain boundary are indicated. The red rectangle shows the extreme probe positions used during the ptychography raster scan (noted as '1', '20', '381' and '400', in agreement with their order within the full scan), while the dashed yellow rectangle corresponds to the area finally imaged with BP. **b** Experimental setup for Bragg ptychography at the synchrotron beamline, detailing the main components used to condition the beam and detect the diffraction signal. The sample frame (**x**, **y**, **z**) and the probe frame (**p**$_1$, **p**$_2$, **p**$_3$) are defined. **c** Example of a diffraction pattern, obtained at the maximum of the Bragg peak, at position 20. **d** A series of four diffraction intensity patterns integrated overall angles along the rocking curve, plotted for the four extreme positions of the raster scan. For (**c**, **d**) the used logarithmic colour scale is indicated.

their potential impact for its use in the design of future fusion reactor components.

## Results

**Data acquisition**. A tungsten, 1% rhenium alloy was manufactured by arc melting, producing a polycrystalline material with grain size of a few hundred microns as determined by scanning electron microscopy (SEM). The addition of rhenium mimics neutron-irradiation-induced transmutation alloying[37]. The material was further implanted with helium ions that modified a ~3 μm thick surface layer (see Methods) and generate neutron-like collision cascade damage[20]. Note that, during fusion reactor operation, helium is generated by transmutation and also diffuses from the plasma. Our use of helium ion implantation effectively mimics these two effects. Finally, using focussed ion beam (FIB) milling, a cross-section sample containing a grain boundary was extracted from the bulk and a region of 15 × 8 μm² was thinned to ~0.45 μm thickness. The resulting sample is shown in Fig. 1a.

BP microscopy was performed at ESRF ID01, a third generation synchrotron beamline, schematically presented in Fig. 1b and described in detail elsewhere[38]. An 8 keV coherent x-ray beam was focused down to about 400 × 200 nm (horizontal versus vertical FWHM of the central lobe in the plane perpendicular to the beam) at the sample position with a set of

Kirkpatrick–Baez (KB) mirrors. The sample was placed on a three-axis translation stage, allowing for nano-positioning and nano-scanning. The stage was mounted on the top of a goniometer cradle, used to orientate the sample in Bragg condition and to map the 3D components of the scattered intensity distribution, by angularly scanning the so-called rocking curve[39]. The sample was scanned across the beam propagation direction, in steps of 100 nm, sufficiently small to collect partially redundant information (corresponding to 75% and 65% overlapping along the two scanning directions, once the footprint elongation of the probe central lobe is taken into account for Bragg geometry, see Method section). For each probe-to-sample position and each angle along the rocking curve, coherent diffraction patterns were recorded on a 2D pixelated detector, placed at an exit angle of twice the Bragg angle, at a distance of 1.4 m from the sample, large enough to ensure far-field regime detection. For our measurements, the (220) specular reflection of the right-hand grain of Fig. 1a was probed. A region of 2 × 2 μm² (enclosed in the red rectangle in Fig. 1a) near the grain boundary and the implantation layer was selected for the ptychography scan, so that both implanted and non-implanted areas could be imaged. Full experimental details are provided in Methods.

As an example representative of the collected data, a 2D coherent diffraction pattern from the He-implanted region is

shown in Fig. 1c. Its asymmetry and extent are strong signatures of the presence of strain and rotation fields within the illuminated sample volume[39]. A set of four patterns are further presented, corresponding to the intensity distribution obtained when integrating the 2D patterns along the rocking curve, for each of the extreme positions of the raster scan (Fig. 1d). Motion of the main peak towards lower Bragg angles, resulting from the strain in the implanted region (i.e. on the 20th and 400th positions), is clearly visible. This suggests that the implantation results in a crystalline lattice swelling, in line with previous reports[20,21]. Moreover, the 3D representation of the intensity distribution allows the investigation of the fringe structure (Supplementary Fig. 1). They arise from the interference from the two surfaces of the sample foil and present a period of about $14\,\mu m^{-1}$. This provides an initial estimation of the sample thickness of ~0.45 μm, in excellent agreement with the thickness expected from SEM.

*3D reconstruction of the sample image.* To go further in the sample structure analysis, the whole dataset is inverted to retrieve the 3D sample crystalline electron density. This requires a detailed modelling of the scattering process: Under the kinematic scattering approximation, the Bragg diffraction intensity distribution, $I_j(\mathbf{q})$, at the $j^{\text{th}}$ probe position $\mathbf{r}_j$ is given by:

$$I_j(\mathbf{q}) = \left| \int_0^\infty P(\mathbf{r} - \mathbf{r}_j)\rho(\mathbf{r})e^{i\mathbf{Q}_{hkl}\cdot\mathbf{u}(\mathbf{r})}e^{i\mathbf{q}\cdot\mathbf{r}}d\mathbf{r} \right|^2, \qquad (1)$$

where $\boldsymbol{q}$ represents the 3D reciprocal space coordinates, and $P$ is the probe function. For the numerical implementation, the direct and reciprocal spaces are introduced. To preserve the data information, the reciprocal frame follows the measurement scanning space $(\mathbf{q}_1, \mathbf{q}_2, \mathbf{q}_3)$. Using a recently developed formalism[40], this sampling corresponds to a direct space frame $(\mathbf{r}_1, \mathbf{r}_2, \mathbf{r}_3)$. Figure 2a shows a schematic representation of both space coordinate frames. These are the spaces in which the inversion is performed before the final object reconstruction is mapped into the orthogonal $(\mathbf{x}, \mathbf{y}, \mathbf{z})$ sample frame (Fig. 1b).

Accessing the phase of the sample scattering function from intensity measurements requires retrieval of the lost phase of the scattered wave field. In forward ptychography, the 3D problem can be decomposed in a series of 2D problems, each of them being solved in the plane that contains the two scanning ptychographic directions[41]. BP inversion is more intricate, because it aims to solve a problem that is intrinsically 3D. Indeed, Eq. (1) cannot be simplified into a series of 2D ptychographic problems, by e.g. separating the probe and the object in the integral. However, only two scanning ptychographic directions are available, transverse to the probe propagation direction (note that along the beam direction, the weak focussing power of x-ray lenses produces an extremely elongated depth of focus). Therefore, while forward ptychography allows the simultaneous retrieval of both probe and sample functions[36], 3D BP requires strong a priori knowledge of the probe.

The structure and size of the probe also have a major impact on the 3D intensity acquisition in BP. To better illustrate this issue, we present the probe used during the experiment (see the cross-section and calculated profile along the beam propagation direction, in Fig. 2b and c). This probe function was characterised prior to our measurement (see Methods). The cross-section presents a rather structured distribution, which includes a central spot of about $400 \times 200\,nm^2$ (intensity FWHM) along the horizontal and vertical planes, respectively and some beam tails and secondary maxima that extend to a much larger area, of about $4 \times 4\,\mu m^2$. These features must be considered when defining the rocking curve angular steps. Indeed the numerical

analysis of the dataset, which is based on the use of the fast Fourier transform, connects reciprocal and direct space through Fourier conjugation relations. Specifically, the sampling angle must be chosen such that the illuminated volume (green line) is fully contained in a cuboid defined in the $(\mathbf{r}_1, \mathbf{r}_2, \mathbf{r}_3)$ reconstruction space (orange line) as shown in Fig. 2a. Depending on the probe structure and on the criterion used to define the probe size, the angular sampling $\Delta\theta$ required along the rocking curve varies substantially, as illustrated in Fig. 2d, for vertical-plane diffraction geometry. This plot presents the sampling condition as a function of the considered (i.e. vertical) probe width W and the sample thickness T, for a Bragg angle $\theta_B$ of 43.9° (see Methods and Eq. (2) for further details). For our sample, the required angular sampling varies from a few 0.01° when only the probe central lobe is accounted for, to about 0.001° when the full probe extent is considered (interestingly, this behaviour is only marginally affected by the sample thickness). While 0.001° angular steps are mechanically accessible at Bragg diffraction optimised beamline set-ups, those small steps are still challenging and result in a detrimental linear increase of the total measurement time. To mitigate issues caused by long data collection, a series of compromises are usually made, such as choosing an angular step only accounting for the central lobe sampling condition and/or fewer ptychographic positions and/or smaller angular range, which reduce the field of view and/or degrade the image quality. Furthermore, the structure of the probe secondary maxima that develop far from the central lobe, i.e. corresponding to high frequencies, is more prone to optics instabilities, limiting the use of prior knowledge of the probe for faithful reconstruction. All this underlines the need to retrieve the full probe distribution function and the sample scattering function simultaneously, a strategy we have implemented and applied in this work to produce high-fidelity maps of lattice strains and rotations.

To succeed with the simultaneous retrieval of the 3D sample scattering function and illumination function, additional information needs to be brought to the inversion problem. Considering the generally small numerical aperture of x-ray optics, which results in a self-similar probe along the beam direction, a straightforward constraint can be derived on the probe invariance. This inversion strategy not only allows recovery of the information in the close vicinity of the central probe lobe, but also of the information arising from much larger distances where only the tails (the series of secondary maxima) of the probe illuminate the sample. According to the sampling principles described above and illustrated in Fig. 2d, this would imply scanning the rocking curve in extremely small angular steps (on the order of a few 0.001°). We circumvented this issue by further up-sampling the rocking curve and retrieving the information in the missing planes (see Methods). This operation corresponds to virtually inserting angular sampling points in-between the measured ones, and to retrieving their diffraction patterns based on the intensity information in the measured ones. Using this strategy, the field of view was increased and the image quality was improved, revealing structural details, discussed hereafter. The whole reconstruction strategy and inversion details are provided in Methods and Supplementary Fig. 2.

*3D analysis of the crystalline lattice distortions.* The iso-surfaces of the retrieved 3D crystalline electron density and the phase $\phi_{220}(\mathbf{r})$ associated with the displacement field $(\phi_{220}(\mathbf{r}) = \mathbf{Q}_{220} \cdot \mathbf{u}(\mathbf{r})$, with $\mathbf{Q}_{220}$ being the Bragg vector for the (220) reflection), are shown in Fig. 3. For the sake of clarity, only the relevant part of the retrieved image is shown, with a mask applied to the all presented maps (see Supplementary Fig. 3 for the mask definition). The presence of a strong edge (seen from the top left corner to the

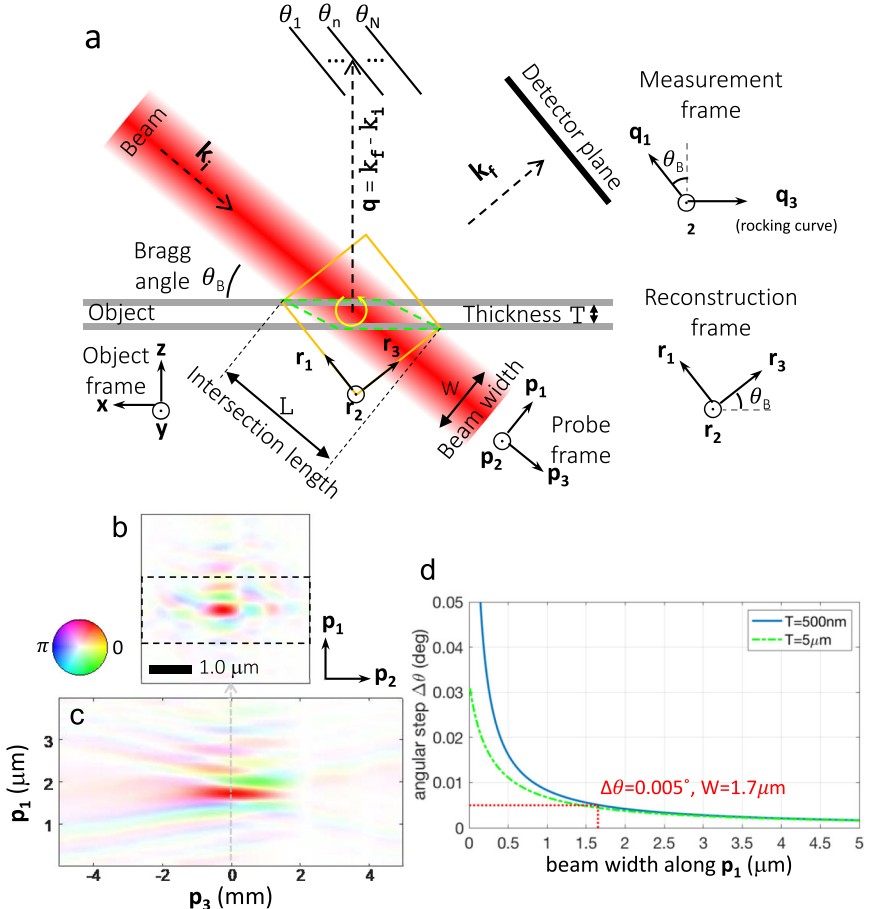

**Fig. 2 Bragg ptychography and sampling principles. a** In the (**x**, **y**, **z**) laboratory frame, a finite size beam illuminates a crystalline thin film sample in Bragg geometry, while a 2D detector collects the coherent diffracted pattern, so that the incident and exit wave vectors (**k**$_i$ and **k**$_f$, respectively) fulfil the Bragg condition. The 3D information is acquired by rotating the sample in the vicinity of the Bragg angle $\theta_B$, along the rocking curve, i.e. at the $\theta_1 ...\theta_N$ positions. The intensity information is recorded as a function of (**q**$_1$, **q**$_2$, and **q**$_3$), the components of the wave-vector transfer **q**. Using a recently developed formalism[40], this sampling depicts a direct space frame (**r**$_1$, **r**$_2$, **r**$_3$). It implies sampling rates along the three directions, that ensure that the illuminated volume (delimited by the green line) is contained within the numerically retrieved direct space (shown as an orange rectangle). The probe frame is also defined as (**p**$_1$, **p**$_2$, **p**$_3$). **b** A typical hard x-ray probe profile presented at the focal plane and (**c**) numerically propagated along the beam axis (see Methods). The dashed rectangle corresponds to the accessible region based on the parameters used during the acquisition of the BP dataset. The full extent of the probe, visible outside this rectangle, is only retrieved through the implementation of the angular up-sampling approach. Along the propagation direction, note the probe invariance over distances as large as a few hundreds of micrometres. The hue rendering colour scale is indicated in (**b**). **d** Sufficiently small sampling steps of $\Delta\theta$ are needed to fulfil the numerical sampling relation. This relation is depicted in the plot, as a function of the probe size W and the sample thickness T.

bottom middle) in the crystalline electron density map corresponds to the expected crystal grain boundary. While the sample density is rather homogenous in most parts of the retrieved field of view, some fluctuations are visible. The pipe-like variations (see black arrows in Fig. 3c and 3d) are well known features, which can be understood when considering the phase map. Here, phase vortices are co-localised with the pipes of missing intensity, a characteristic feature of dislocations in noise-limited BCDI data[27,42–44]. Note that the observations of their sharp structure confirms the good quality of the dataset including possible beam-to-sample position uncertainties. Similarly, additional density oscillations appear in regions, where the phase wraps rapidly, e.g. at the top right corner[34]. A few density voids are observed in the vicinity of the grain boundary, corresponding to cavities in the crystal. Finally, at the edges of the raster scan, where both the counting statistics and overlap constraint are reduced the reconstruction quality is degraded. The probe function retrieved simultaneously from the BP dataset (cross-section profile shown in Fig. 3e) exhibits the expected characteristic features (e.g. the

central lobe with size of $450 \times 190$ nm$^2$, intensity FWHM), in excellent agreement with the profile obtained separately and previously shown in Fig. 2b. The missing triangular region at the bottom left of the probe reconstruction is due to the limited extent of the crystal set by the grain boundary. Note the extent of the field of views, of about $0.6 \times 6 \times 6$ µm$^3$ (**z** × **y** × **x**) for the 3D sample and $5.1 \times 3.9$ µm$^2$ (**p**$_1$ × **p**$_2$) for the probe cross-section. The probe field of view is therefore large enough to cover most of the probe extent, including the central lobe and the tails (as shown in Fig. 2b). If the intensity information were not limited by the grain boundary (non-scattering crystal), we estimate the field of view along **x** could be as large as 7.8 µm. Finally, the spatial resolution of this 3D reconstruction is estimated as $37 \times 40 \times 39$ nm$^3$ along **z**, **y** and **x** respectively (see Supplementary Fig. 4).

As a matter of comparison, the same dataset was reconstructed using the formerly employed approach[29,33], i.e. no angular up-sampling and using a fixed probe characterised by transmission ptychography. The 3D reconstruction numerical volume was

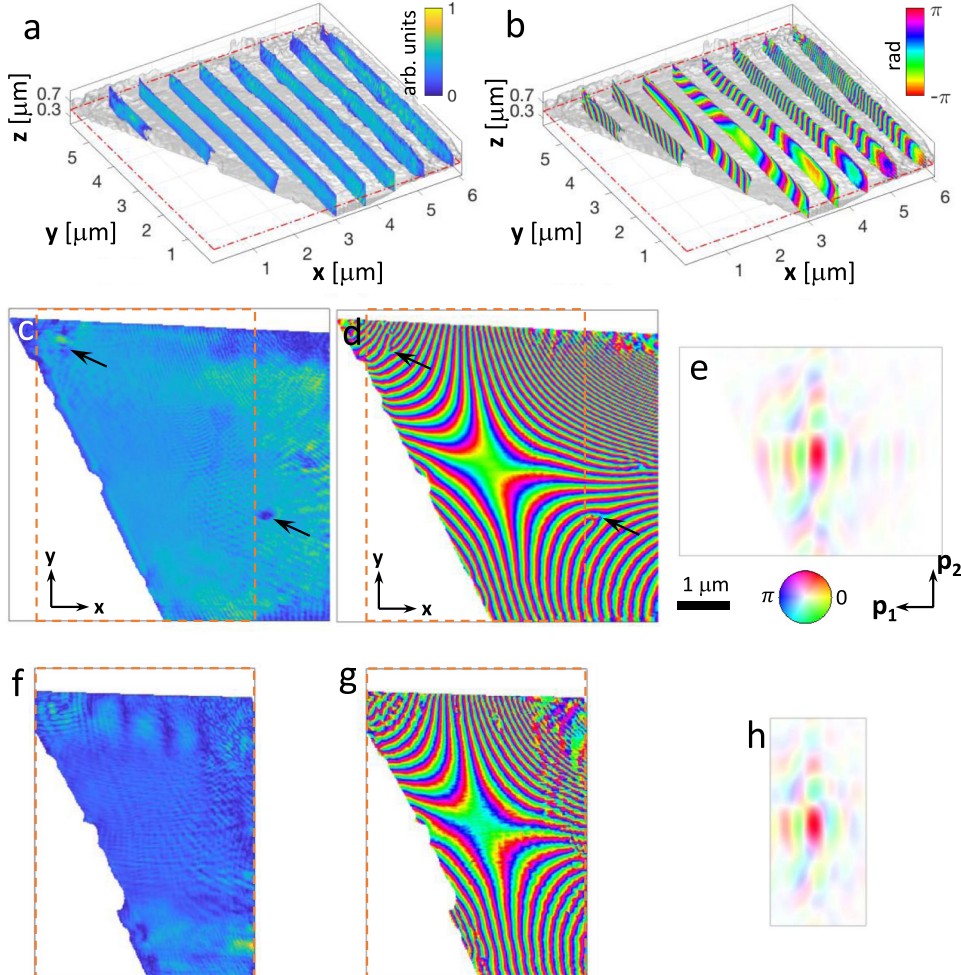

**Fig. 3 Extended field of view retrieved image. a** 3D density and (**b**) phase plots, retrieved with the proposed BP approach in the (**x**, **y**, **z**) laboratory frame. The grey volume represents an iso-surface based on the recovered object density, while the slices show the internal structure of density and phase, respectively. The field of view extends over $0.6 \times 6 \times 6 \ \mu m^3$. **c, d** Cross sections of the crystal density and phase, respectively, shown over the plane indicated by red dashed rectangles at $\mathbf{z} = 4.7 \ \mu m$ in (**a**) and (**b**), and (**e**) the associated retrieved probe. **f, g** 3D density and phase reconstructions, respectively, obtained from the same dataset, using the former BP approach. Note the strong reduction of the field of view (evidenced by the dashed rectangle) and the degradation of the image quality. A comparison of the total possible field of view is shown in supplementary Fig. 3d, including regions of the possible field of view that are non-scattering due to the finite extent of the considered crystal. **h** Probe cross-section, plotted in the ($p_1$, $p_2$) plane, used for this second reconstruction, limited along $p_1$ according to the conjugation relations applied to the experimental parameters. As the probe is kept fixed during this process, the probe profile is extracted from a separate and dedicated probe reconstruction (Methods). The colour scale for the density, phase and probe (hue rendering) are indicated on the plots (**a**), (**b**) and (**e**), respectively.

directly designed according to the conjugation relation applied to the experiment parameters, in particular the angular step. The final results are shown in Fig. 3f–h. The reduction in the field of view is particularly evident along the **x** direction (maximum field of view of 4.6 μm compared to 7.8 μm for the angular up-sampled approach), which is the direction mostly impacted by the angular sampling along the rocking curve. Although the main focal spot is fully contained in the probe function, the beam tails are clearly truncated. This impacts the quality of the object image, where the grain boundary is still visible, but artefacts, such as fluctuations in both crystalline electron density and phase, are more abundant. The previously evident dislocations are barely visible. This direct comparison underlines the merit of the improved inversion strategy we have proposed.

The high-quality extended reconstruction was used to analyse the structural features present in the implanted crystal. To this end, we used the 3D phase map (an unwrapped version of it being converted into the displacement field component and presented in

Supplementary Fig. 5) to extract the lattice strain projected along $\mathbf{Q}_{220}$ and the lattice rotations about the **x** and **y** axes, hereafter referred to as $\varepsilon_{zz}$, $\omega_x$ and $\omega_y$, respectively (see Methods). The full field-of-view strain and tilts are presented in Fig. 4, while specific details are highlighted in the extracted profiles shown in Fig. 5. The top edge of the reconstructed volume (Fig. 4) corresponds to the former sample surface during ion-implantation (note that, the strain increase observed at the very top surface is an artefact caused by aliasing due to the probe extent). The 3D strain map (Fig. 4a and b) clearly shows the He-implantation induced strain of about $3 \times 10^{-4}$, evident to a depth of approximately 2.8 μm (Fig. 5b, c). In the vicinity of the two FIB-processed surfaces, large positive strains, corresponding to a lattice expansion, can be seen. These strains extend to similar depths (about 150 nm for the top surface and 100 nm for the bottom surface) and have similar magnitude (~$1.5 \times 10^{-3}$ and ~$1.1 \times 10^{-3}$ for the top and bottom surfaces, respectively). Several dislocations are also observed in both implanted and non-implanted layers (highlighted in Fig. 4

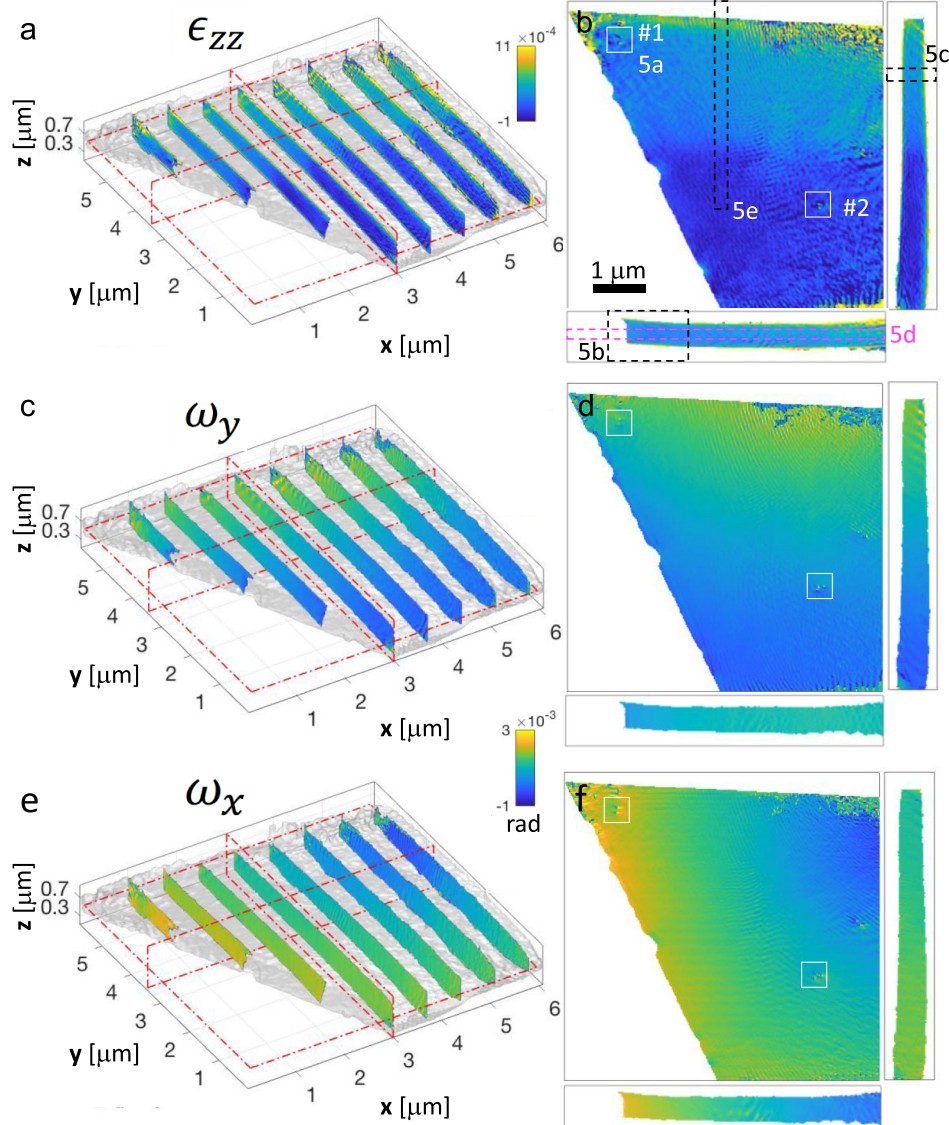

**Fig. 4 Strain and tilts revealed in the He-implanted polycrystalline tungsten foil. a** 3D iso-surface density plot overlaid with the $\epsilon_{zz}$ strain map, plotted in the (**x**, **y**, **z**) laboratory frame. Three planes used for further investigations are indicated as red rectangles. **b** Cross-sections of the $\epsilon_{zz}$ map extracted over the planes indicated in (**a**). **c** 3D iso-surface density plot overlaid with the $\omega_y$ lattice rotation around **y**-axis. **d** Cross-sections of $\omega_y$ extracted over the planes indicated in (**c**). **e, f** Same as (**c, d**) for the $\omega_x$ lattice rotation around **x**-axis. All scale bars and angular colour scales in radian are indicated on the plots.

and analysed in more details in Fig. 5a). Besides these, $\epsilon_{zz}$ presents a rather homogeneous behaviour in the whole implanted layer (Fig. 5c–f). We encourage the reader to view the Supplementary Videos as these provide an interactive way of visualising the multi-dimensional dataset. Supplementary Video 1 shows the object iso-surface, whilst Supplementary Video 2 and Supplementary Video 3 each show a perpendicular plane propagating through the object with the $\epsilon_{zz}$, $\omega_x$ and $\omega_y$ components displayed. Note that the access to the full set of strain and tilt components would require the investigation of at least three non-orthogonal Bragg reflections, a technically challenging though conceivable experiment.

*Comparison with other crystalline sensitive approaches.* To further illustrate the interest of our BP result, characterisation using more routinely applied strain microscopy methods, namely high-resolution electron back scattering diffraction (HR-EBSD) and x-ray micro-beam Laue diffraction measurements, were

performed for the same region of the sample. These results are shown in the Supplementary Fig. 6. Whilst these methods are considered a mainstay for the analysis of strain and tilt distributions in polycrystalline materials, their limitations are obvious in the present case. HR-EBSD presents a lateral spatial resolution of ~100 nm[45] and is only surface sensitive (down to a depth of about 10 s of nm) due to the short mean free path of backscattered electrons[46]. On the other hand, micro-beam Laue diffraction is bulk sensitive, but the obtained 2D maps represent the integration of the strain and tilt information through the whole sample thickness. Its transverse spatial resolution, which integrates the size of the focused beam, the scanning precision and the step size, is approximately 0.5 μm in the present case. Some indication of positive lattice strain along the upper edge of the sample (where the implanted layer is known to be) was recorded by HR-EBSD and micro-beam Laue diffraction. How-ever, neither approaches were able to definitively capture the

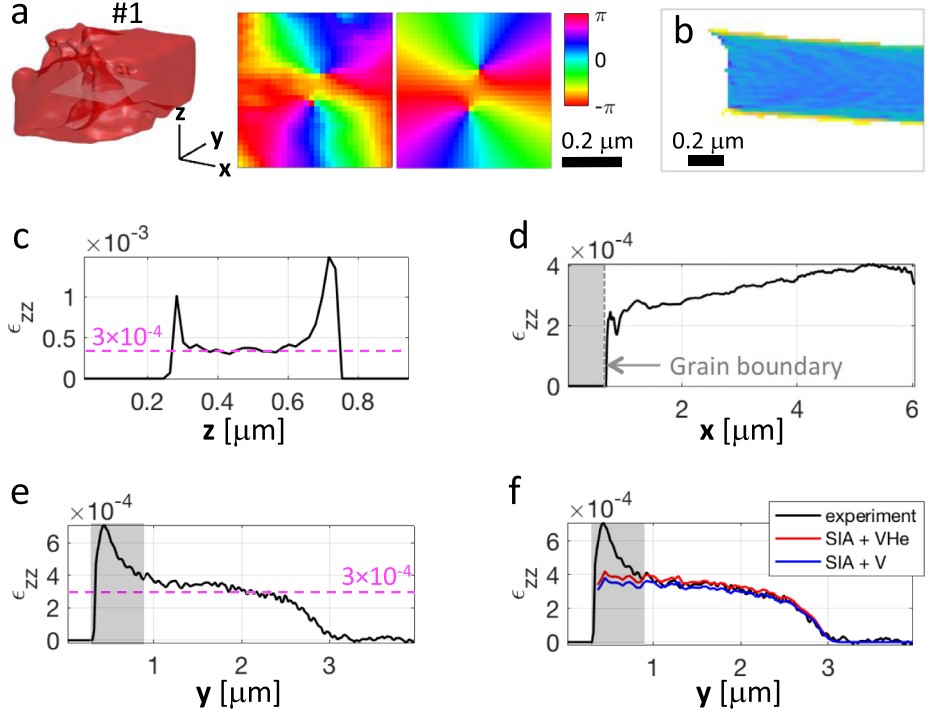

**Fig. 5 Extracting displacement fields and strain profiles. a** Left: zoomed-in 3D iso-surface density plots of the dislocation #1 highlighted by a white rectangle in the implanted layer, shown in Fig. 4b, d and f and displayed in the (**x**, **y**, **z**) laboratory frame. Middle: 2D cross-section map of the $\phi_{220}$ reconstructed phase. Right: estimated phase variation resulting from simulation (see Supplementary Fig. 8). **b** Zoomed-in view of the $\varepsilon_{zz}$ strain map extracted from the implanted region in the vicinity of the grain boundary (as shown in Fig. 5b). **c**, **d**, **e** One-dimensional cross sections of the average $\varepsilon_{zz}$ strain map, across the film thickness, perpendicular to the grain boundary within the implanted region and along the implantation direction, respectively, as indicated in Fig. 4b. In (**e**) the grey area corresponds to a region where $\varepsilon_{zz}$ is slightly degraded due to a bit of parasitic aliasing along the **y** direction. **f** Calculated strain profile along the implantation direction, assuming the damage microstructure consists of Frenkel pairs with vacancy filled by 1 He atom (SIA + VHe) or as Frenkel pairs alone (SIA + V). See Supplementary Fig. 9 for details. All scale bars and angular colour scales in radian are indicated on the plots.

presence of the implanted layer with a high level of clarity or any indication of the dislocations and the fine details of the strains revealed by BP.

## Discussion

The success of our approach relies on the ability of our proposed method to retrieve the far-field information in planes, whose intensity distributions were not experimentally measured. Of course, this process has a limit and a dedicated analytical work is under way to further derive this mathematically. It was numerically determined for this experiment (Supplementary Fig. 7). Qualitatively, the limit can be understood by considering that the numerical aperture of the focusing optic broadens the signal along the corresponding directions in the reciprocal space, in particular along the rocking curve direction. Therefore, the intensity information obtained in a given detector plane carries additional features arising from the broadening of the signal along the direction perpendicular to the detector plane. Interestingly, we observed that the reconstruction quality using 7 angular measurements (up-sampling ratio of 18 and angular step of 0.03°) is comparable to the quality of the image produced by the former reconstruction strategy, using 42 angles. From this comparison, considering the gains with respect to the measurement time (×6) and to the achieved field of view (×1.4 considering the limits imposed by the crystalline grain size or ×1.7 considering the whole accessible field of view), a total gain of about 8–10 is estimated. This significant improvement brought by

our approach should further benefit from the advent of 4th generation synchrotron sources and their expected gain (of about 100) in coherence flux.

Regarding the crystalline properties imaged in the sample, we note that the 3D lattice strain and rotation maps obtained using our approach are not accessible in any other way and reveal important features. In the vicinity of the top and bottom sample surfaces, large positive strains, corresponding to a lattice expansion, can be seen. The strains extend to similar depths (~120 nm), have similar values (~1.1–1.5 × 10$^{-3}$) and are present over the entire sample surfaces. Their spatial distribution and location are consistent with the effects of residual FIB-induced damage. FIB imaging and machining are known to cause dramatic material changes such as introduction of lattice defects[47], large lattice strains[48], amorphization[49], and formation of Ga intermetallics[50]. In electron microscopy, FIB damage is often indistinguishable from the intrinsic defects and damage features of interest in the sample[51,52]. In the present case, it is worth noting that FIB-induced damage remains after employing preventative and mitigating measures, routinely employed in fabricating strain microscopy samples for electron microscopy including sacrificial capping, glancing incidence ion milling and the removal of surface material via low energy polishing (see Methods for fabrication details). We observe that even these residual effects lead to large lattice distortions. However, the beauty of our approach is that sample regions affected by FIB damage can be unambiguously identified and then excluded from further analysis. This strain distribution underlines the importance of probing the 3D

information, a strategy that is highly challenging with electron-diffraction based microscopy approaches. While electron microscopy is able to provide strain maps with sensitivity comparable to the one of X-ray Bragg diffraction (about $10^{-4}$), the 2D electron microscopy image only provides strain information averaged over the sample thickness direction. Our 3D approach dramatically simplifies data interpretation and gives certainty that features observed in the sample are not artefacts of the preparation. In the following only sample volumes more than 150 nm from sample surfaces, and thereby unaffected by FIB, are considered.

The $\varepsilon_{zz}$ lattice strain map (Fig. 4a and 4b) shows substantial strain with an average value of $3 \times 10^{-4}$ in the helium-implanted layer. This is in good agreement with low-resolution micro-beam Laue measurements of helium-implantation-induced lattice strain[21], estimating strains of about $2.4 \times 10^{-4}$. This lattice expansion can be understood in terms of Frenkel defects where He occupies the vacancy, preventing recombination[20,53,54]. Atomistic simulations have suggested that the He-occupied vacancy and self-interstitial atom (SIA) form a stable, bound configuration[53,54]. This means that, at room temperature, the SIAs are not free to move and cannot cluster. The net positive lattice swelling arises as self-interstitials have large positive relaxation volume ($\Omega_r$ (SIA) = 1.68), while vacancies, with or without He, have smaller negative relaxation volume ($\Omega_r$(V) = −0.37, $\Omega_r$ (V + He) = −0.24)[20]. The lower bound Frenkel pair density, required to produce the measured strain, can thus be estimated as ~700 appm (see Supplementary Fig. 8 for details). Comparing to the damage predicted by binary collision simulations (Supplementary Fig. 8), this suggests only ~4% of the induced irradiation damage is retained. Using this retention efficiency, the lattice strain profile anticipated from the depth-variation of damage in binary collision simulations can be predicted (Fig. 5f). The agreement with the measured strain profile is quite remarkable, indicating that, at least on the scale of our measurements, there is no appreciable defect migration. Noticeably, this strain can be distinguished from that of larger dislocations so that strain from dislocations can be excluded from the consideration of strain due to invisible defects. Whilst there is a clear change in lattice strain from the He-implanted layer to the un-implanted material beneath, there is no discernible, abrupt change in lattice rotation at the boundary between the implanted and un-implanted material (Fig. 4c–f). This is a very important evidence, since, in principle, a shear deformation of the lattice during ion implantation could lead to rotations about the **x** and **z** axes. Such a shearing would indicate the preferential formation of defects with specific orientation[55]. The lack of sharp change in lattice orientation at the un-implanted/implanted material interface suggests that defects are randomly oriented, leading to a purely volumetric strain[55,56]. This important result should greatly simplify the simulation of irradiation-induced strain in reactors[56], which is anticipated to be one of the main mechanisms driving in-service degradation of fusion reactor armour.

Across the implanted layer, the homogeneous behaviour of $\varepsilon_{zz}$ indicates a uniform distribution of defects, at least within the resolution and sensitivity limit of the present measurements. Previously, the formation of a zone denuded of large irradiation-induced dislocations loops was reported close to grain boundaries (within 20–50 nm) in self-ion irradiated material[57]. Our results suggest that for small defects and defect clusters, such as those created by helium ion-irradiation[20], this is not exactly the case. From a detailed fit of the strain profile, shown in Supplementary Fig. 9, we see evidence of only a partially defect-denuded region, corresponding to a partial release of the strain down to about $2.07 \times 10^{-4} \pm 0.13 \times 10^{-4}$ over a thickness of about $72 \pm 8$ nm (here, the error bars are the standard deviations arising from the spatial variations observed in the strain profile at

different locations). One possibility is that defect size affects the degree of denuding, whereby small defects, which experience smaller elastic driving forces to sinks, lead to partially denuded zones. In addition, the high migration energy of He-filled vacancies[58,59] may restrict the growth of denuded zones. Either way, our observations suggest that the potential of grain size reduction as a means of reducing irradiation defect accumulation may be limited.

Finally, the quality of our crystalline image allows us to consider in more detail the dislocations visible in the reconstructed sample. An iso-surface rendering of the crystalline electron density, as well as the associated phase variation in the sample mid-plane are shown in Fig. 5a and Supplementary Fig. 10 for dislocation #1 (implanted layer) and #2 (un-implanted region), respectively. Dislocations in tungsten have predominantly **b** = 1/2 ⟨111⟩ Burgers vector, although dislocations with **b** = ⟨100⟩ have also been reported[44]. For a given Bragg reflection, only dislocations for which $\mathbf{Q}_{hkl}\cdot\mathbf{b}$ is non-zero are visible[60]. For the present (220) reflection, this means that the observed dislocations may have the following Burgers vectors: 1/2[111], 1/2[11-1], [100] or [010]. Dislocation #2 shows a central pipe of missing intensity[26]. The phase variation around the dislocation line shows a vortex with a total phase increase of $4\pi$. The phase variation agrees well with the simulated phase for a dislocation in the thickness direction, as shown in Supplementary Fig. 10. In contrast, dislocation #1 shows a more complex and surprising structure: Here two separate pipes of reduced intensity are clearly identified, suggesting that in fact this dislocation corresponds to two parallel dislocations, each with an associated phase vortex of $2\pi$. This is the signature expected from two partial dislocations linked by a stacking fault on the (-110) plane. The presence of dissociated dislocations in tungsten is surprising since bcc metals have comparatively high stacking fault energy[61]. However, recent ab-initio calculations show that in fact, the addition of Re can substantially reduce the stacking fault energy in tungsten, a scenario possibly at play here[61].

In summary, thanks to the development of an upgraded BP approach, we obtain quantitative 3D maps of nano-scale lattice strain and rotation in a He-implanted Tungsten crystal. Our inversion scheme provides an improved sample image quality, not only because the probe and sample can now be accurately disentangled, but also because the signals arising from regions far from the probe central lobe are now accounted for. These maps, of high quality and large extent, reveal numerous, otherwise inaccessible, crystalline features. Beyond a FIB-damaged layer, which can now be unambiguously discarded from the analysis, we observed strains and lattice rotations caused by 'invisible' helium-implantation-induced defects, identified to be of random orientation. Surprisingly, we only found a partially defect-denuded region in the vicinity of the crystal boundary. These results provide new insights, essential for predicting the effects of ion-irradiation on metals. Moreover, they pave the way for highly detailed investigation of complex next-generation crystalline materials, e.g. refractory high-entropy alloys for extreme environments[62,63].

## Methods

**Sample preparation.** The sample was produced from a bulk polycrystalline tungsten-1wt% rhenium alloy, manufactured by arc melting from high purity elemental powders. The sample surface was mechanically ground and further polished using diamond paste. A final chemo-mechanical polishing step (with 0.1 μm-colloidal silica suspension) was used to produce a high-quality surface finish. The grain size in the polycrystal was ~500 μm. The sample was implanted with helium ions to produce a 2.8 μm thick implanted layer. To achieve a relatively uniform damage level (0.02 ± 0.003 dpa between 0 μm and 2.8 μm depth) and injected helium concentration (310 ± 30 appm between 0 μm and 2.8 μm depth), implantation was performed using a number of different ion energies up to 1.8 MeV at the Ion Beam Centre, University of Surrey. The ion energies and corresponding fluence are provided in Supplementary Table 1. For BP

measurements, a small specimen (~25 × 14 × 0.5 µm³) was extracted from the bulk using a focused ion beam (FIB) lift-out process. This preparation method is adapted from that developed for the preparation of isolated micro-crystals for BCDI[44]. To minimise the effects of lattice damage induced during the lift-out process a combination of protective capping, glancing incidence milling, and low energy polishing was used. Specifically, the surface of the bulk sample was first covered with carbonaceous platinum, to protect the tungsten from normal incidence angle milling ions, using an electron-beam, which does not introduce any damage in the implanted material. This produces a thin layer of about 0.1–0.2 µm, which appears as a dark line in the SEM picture (Fig. 1a). On top of this layer, 2 µm of carbonaceous platinum were further deposited using the gallium-ion-beam, which achieves a balance of sputtering and deposition. This is faster than the electron-beam assisted deposition but more intrusive, justifying the prior deposition of the electron-beam assisted deposition layer. The gallium-deposited platinum appears with a different contrast at the top of the sample. Trenches on either side of the lift-out sample were made using FIB. The sample was then undercut and attached to a micro-manipulator. The sample was finally attached to a copper TEM-lift-out holder at 90 degrees to its original orientation i.e. the walls of the trench became the top and bottom surfaces of the sample for all measurements. As the sample was still rather thick at this point (1–2 µm), FIB was used to further thin down the sample. This was performed at 30 keV, with a current of ~240 pA or slightly lower. Finally, the obtained lamella was polished with a 2 keV gallium ion beam to remove most of the damage from the previous FIB steps. It results in a ~0.45 µm thinned region around the implantation layer region, in the vicinity of the grain boundary. The implanted layer starts from the dark line downwards (Fig. 1a). The (220) lattice planes of the interested grain (further away from the copper support end in Fig. 1a) are parallel to the two lift-out surfaces.

**Experimental details—Bragg ptychography.** The x-ray beam of the ID01 beamline (ESRF) was passed through a Si-111 double crystal monochromator to provide a monochromatic beam with energy of 8 keV (or wavelength λ of 1.54 Å) with a bandwidth of about $1 × 10^{-4}$. Kirkpatrick–Baez mirrors were used to focus the beam onto the sample, horizontally (H) and vertically (V). A pair of slits placed upstream of the mirrors were set to a width of 60 × 200 µm² (H × V), selecting a spatially coherent beam for the experiment. From a dedicated (forward ptychography) beam characterisation, the beam profile was extracted and the central spot size (intensity FWHM) was measured as 400 × 200 nm² (H × V). The resulting numerical aperture (NA) was estimated to be $1.9 × 10^{-4}$ and $3.7 × 10^{-4}$ (H × V) using the extent of the calculated over-focused beam and the depth of focus of 1950 × 540 µm² (H × V). More details are given in the following Methods section.

For the BP measurements, the specular (220) Bragg reflection of the grain away from the copper support was chosen. It corresponds to a Bragg angle $θ_B$ of 43.9°. This inclined geometry results in an elongated footprint of the central beam spot with a size of 400 × 288 nm² (H × V). The BP acquisition was performed after ensuring the thermal stability of the experimental hutch was reached and assessing the mechanical stability and repeatability of the set-up. The intensity patterns were measured with an area detector (Maxipix, 516 × 516 pixels, pitch size of 55 × 55 µm²) located at 1.4 m from the sample. A total of 42 angular positions, with an angular step size of 0.005°, were taken along the rocking curve. At each of these angles, a set of 20 × 20 positions, with a step size of 100 nm, was used for the raster scan. This corresponds to an area of 2 × 2 µm². The exposure time per frame was set to 0.2 s.

Immediately after BP measurements, the experimental geometry was changed to the forward geometry to perform a conventional transmission ptychographic scan with a Siemens star pattern, in order to retrieve the illumination profile[36]. An Archimedean spiral scan of 513 positions, with a step size of 100 nm, was performed and for each position the detector was exposed for 1 s to measure the diffraction pattern. A sub-region of 334 × 334 pixels, centered around the forward beam, was used to retrieve the probe profile at the focal plane using standard approaches based on the extended ptychographical iterative engine algorithm (ePIE)[64]. The initial guess for the probe was a simulated beam profile based on the experimental setup, i.e. calculated via the Fourier transform of a rectangle that is the numerical aperture of the beam. The initial guess for the object was simply a uniform matrix of 1 s. The gap between the different sensor modules was masked and left free. The reconstruction was run for 100 iterations, where the difference between the measured diffraction patterns and the calculated ones is very small and further iterations provide very little further convergence.

**Angular sampling along the rocking curve for 3D fourier transform based Bragg coherent diffraction imaging.** Due to the use of a 3D discrete Fourier transform to describe the propagation between real and reciprocal spaces, the angular sampling pitch $Δθ$ and the window size $R_3$ of the illumination function along $r_3$ are linked via $Δθ = λ/2R_3 \sin(θ_B)\cos(θ_B)$, according to the sampling relations[40,65], where, considering the geometry depicted in Fig. 2a, $R_3 = T\cos(2θ_B)/\sin(θ_B) + W/\tan(θ_B)$. Here T is the sample thickness and W is the beam width along $p_1$. This leads to the following relation between the beam width W and the sampling pitch $Δθ$:

$$W = λ/2Δθ[\cos(θ_B)]^2 - T\cos(2θ_B)/\cos(θ_B) \tag{2}$$

For the used angular step size of 0.005° with a sample thickness of 500 nm, W is about 1.7 µm. It is big enough to encompass the focal spot, but not the secondary maxima in the probe tails. Those strong beam tails, which extend beyond the window size, result in an aliasing artefact according to the Nyquist sampling theorem[66]. Therefore, with this angular step size, the quality of the reconstruction is limited.

**Reconstruction strategy.** The overall reconstruction strategy stems from previous works in 3D BP[28,34,65] with some pivotal adaptations that are required to obtain the high quality, stable reconstruction shown in Fig. 3. The reconstruction is performed via the following cost-function to be minimised over all the probe positions

$$\mathcal{L}(ρ, P, b) = \sum_{j=1}^{J} \mathcal{L}_j(ρ, P, b) \tag{3}$$

where ρ, P and b, respectively, are the 3D electronic density (i.e. the sample), the 3D probe function and the incoherent intensity background. Individual cost-functions are defined by[28,34]

$$\mathcal{L}_j(ρ, P, b_j) := \left\| ω(\mathbf{q}) × \left( I_j^{1/2}(\mathbf{q}) - h_j^{1/2}(\mathbf{q}) \right) \right\|^2 + μ_j \sum_{\mathbf{r} \in \bar{S}} |ρ(\mathbf{r})|^2 \tag{4}$$

where $I_j(\mathbf{q})$ and $h_j(\mathbf{q}) = |\Psi_j(\mathbf{q})|^2 + b(\mathbf{q})$ are the measured and predicted photon counts for the j-th probe position, respectively and $\Psi_j(\mathbf{q})$ is the Fourier transform of the exit wave field. The detector mask $ω(\mathbf{q})$ is designed to discard hot or dead camera pixels but also to provide the expected angular up-sampling factor (described in detail below). The second, quadratic term is a thickness-support regularisation applied to the retrieved sample preventing the density of the sample to build-up in $\bar{S}$, the set of points *outside* the sample support. The regularisation parameters $μ_j ≥ 0$ adjust how the final solution should comply with this support constraint; in the Bragg geometry, we note that this regularisation is pivotal in dealing with the intrinsic lack of diversity of the beam along the propagation direction[65]. The reconstruction strategy aims to simultaneously retrieve ρ, P and b from the set of measurements $\{I_j(\mathbf{q})\}_{j=1}^{J}$ via the minimisation of Eq. (3).

*Presentation of the reconstruction algorithm.* The reconstruction algorithm was derived from the ePIE[64] approach, with specific adaptations that allow the probe function to be accurately retrieved in the Bragg geometry. More specifically, a probe and object updates associated with the current (updated) position $j ∈ \{1 \cdots J\}$ are given by

$$ρ(\mathbf{r}) \leftarrow ρ(\mathbf{r}) + λ_ρ^{-1}(\mathbf{r}) × ∂_{ρ;j}(\mathbf{r}) \tag{5}$$

$$P(\mathbf{r}) \leftarrow \mathcal{B}_{k_i} \mathcal{R}_{k_i}(P(\mathbf{r}) + λ_P^{-1}(\mathbf{r}) × ∂_{P;j}(\mathbf{r})) \tag{6}$$

where $∂_{ρ;j}$ and $∂_{P;j}$ are the gradients of $\mathcal{L}_j$ with respect to ρ and P, respectively

$$∂_{ρ;j}(\mathbf{r}) = P^*(\mathbf{r} - \mathbf{r}_j)[ψ_j'(\mathbf{r}) - ψ_j(\mathbf{r})] - μ_j[1 - S(\mathbf{r})]ρ(\mathbf{r}) \tag{7}$$

$$∂_{P;j}(\mathbf{r}) = ρ^*(\mathbf{r})[ψ_j'(\mathbf{r}) - ψ_j(\mathbf{r})]. \tag{8}$$

The projection and back-projection operators $\mathcal{R}_{k_i}$ and $\mathcal{B}_{k_i}$, respectively, act altogether along the direction of the incoming beam $\mathbf{k}_i$; the pair of operators basically enforces, in the probe update (Eq. 6), the invariance of the probe along the direction $\mathbf{k}_i$ (described in details below). The scaling factors $λ_ρ^{-1}$ and $λ_P^{-1}$ in Eqs. (5) and (6) aim to provide some convergence acceleration in the joint update[67]

$$λ_{ρ;j}(\mathbf{r}) := (1 - β)|P(\mathbf{r} - \mathbf{r}_j)|^2 + β|P(\mathbf{r} - \mathbf{r}_j)|^2_{max} + μ_j(1 - S(\mathbf{r})) \tag{9}$$

$$λ_{P;j}(\mathbf{r}) := (1 - α)|ρ(\mathbf{r})|^2 + α|ρ(\mathbf{r})|^2_{max} \tag{10}$$

where $α ≤ 1, β ≥ 0$ are constant parameters whose tuning is left to the user. In Eqs. (7), (8) $ψ_j(\mathbf{r}) = P(\mathbf{r} - \mathbf{r}_j)ρ(\mathbf{r})$ is the exit wave-field and $ψ_j'(\mathbf{r})$ is the updated/corrected exit wave-field, whose Fourier transform is denoted by $\Psi_j'(\mathbf{q})$, and the support function $S(\mathbf{r})$ vanishes whenever $\mathbf{r} ∈ \bar{S}$ and is equal to one otherwise. Finally, the incoherent intensity background $b(\mathbf{q})$ is jointly estimated, using a multiplicative update that enforces the positivity of the background (as long as the initial estimate is positive) and reads for $j ∈ \{1 \cdots J\}$

$$b(\mathbf{q}) \leftarrow b(\mathbf{q}) × \left( (1 - γ) + γ \left[ \frac{I_j(\mathbf{q})}{h_j(\mathbf{q})} \right]^{1/2} \right)^2 \tag{11}$$

with $γ ≥ 0$. This background helps to account for parasitic scattering and small instabilities of the set-up. The updating relations Eqs. (5), (6) and (11), are at the core of the iterative joint reconstruction strategy used in this work.

*Probe retrieval: invariance property and spatial extension.* The invariance of the probe function along the incident beam direction $\mathbf{k}_i$ is pivotal as it greatly reduces the solution space for the probe and thereby enables the actual, simultaneous

reconstruction of the probe and the object. To justify this assumption, we note that the typical NA for a focusing optic is a few $10^{-4}$ at hard x-ray energies, hence resulting in typical depth-of-focus (DOF) of a few 100 μm, as given by the relation[68] $DOF = \lambda/(2NA^2)$. In BP applications, the maximal thickness of the sample is about 2 μm, a limit set by the longitudinal coherence length of the x-ray source. Along the beam direction, the length L of the probe-object intersection volume, enclosed by the dashed green parallelogram in Fig. 2a, is $L = T/\sin(\theta_B) + W/\tan(\theta_B)$, which is on the order of a few microns. At this scale, the probe invariance along its propagation direction can be safely assumed. In each iteration, this beam propagation constraint is enforced in the current probe estimate, by the projection/back-projection operator pair shown in Eq. (6).

The spatial extension of the probe is directly connected to the angular sampling step-size $\Delta\theta$. From the numerical point of view, as shown in Fig. 2a, this step-size is driven by the window size in the reconstruction frame (i.e. the orange rectangle, determined by the sampling in the measurements), which has to be bigger than the intersection volume (the green lines), see Fig. 2a. Therefore, using Eq. (2), the angular step $\Delta\theta$ must meet the following requirement:

$$\Delta\theta \leq \lambda/(2\cos(\theta_B)[T \cdot \cos(2\theta_B) + W \cdot \cos(\theta_B)]) \quad (12)$$

However, in practice, the angular step-size can be designed so that it is bigger than the sampling requirement $\Delta\theta$ along the rocking curve, without losing information. In this case, to preserve the real to reciprocal space conjugation relation, one can numerically insert virtual angular sampling points between the measured angles to access an effective angular step that meets the Nyquist sampling requirement. Only the measured angles are then used for the intensity constraint during the reconstruction, and the virtual ones are left free. Using this approach, one can build a corrected field $\Psi'_j$ for all diffraction planes, from the following equation[65]

$$\Psi'_j(\mathbf{q}) = [|\Psi_j(\mathbf{q})| + \omega(\mathbf{q}) \times (I_j^{1/2}(\mathbf{q}) - |\Psi_j(\mathbf{q})|)] \frac{\Psi_j(\mathbf{q})}{h_j^{1/2}(\mathbf{q})} \quad (13)$$

where the mask function $\omega(\mathbf{q})$ is set to 1 for the measured angles and 0 for the virtual angles, i.e. the algorithm is left to retrieve the diffracted field for those virtual angles. In our dataset, the angular up-sampling implemented via Eq. (13) was used, with an angular up-sampling ratio set to 3, which corresponds to the insertion of two virtual angles between every two angular measurements within the reconstructed dataset. This gives an effective angular step of 0.0017° and an effective window size of 5 μm for the probe reconstruction along $\mathbf{p}_1$. From a computational viewpoint, we note that the updates (Eqs. (5), (6) and (11)) were implemented via a modified 3D Fourier transform recently developed[40,69]. This transform allows the unknown quantities to be retrieved in an orthogonal frame determined by the detector plane and the exit beam direction $\mathbf{k}_f$ while still preserving the statistics of the measured signal, i.e. there is no need to interpolate the measurements acquired in a non-orthogonal frame (see Fig. 2a).

*Inversion parameters.* For the reconstructions, a sub-region of $200 \times 160$ pixels, centered around the Bragg peak, was cropped out from the full-size measurements. In the reconstruction frame, it results in pixel sizes of $13.6 \times 24.6 \times 42.3$ nm³ ($\mathbf{r}_1 \times \mathbf{r}_2 \times \mathbf{r}_3$). When converted into the object frame, the pixel size was equalised to 18.9 nm (i.e. the pixel size along z). In the detector plane, the gap between different sensor modules was masked out. We set the support-thickness regularisation to $\mu_j := \mu \times |P(\mathbf{r} - \mathbf{r}_j)|^2_{max}$ with $\mu = 0.02$ and a first set of 1000 updates were performed with Eqs. (5), (6) and (11), where we set $\alpha = 1$, $\beta = 0.5$ and $\gamma = 0.1$. The initial guess was a flat field with a constant value of 0.1 for the background, and a pre-characterised illumination obtained from forward ptychography for the probe; we note that this initial probe is not a necessity for the convergence of the joint reconstruction (alternatively, other illumination functions were used as an initial guesses and successful reconstructions could be produced, showing the robustness of the inversion process, see Supplementary Fig. 2). As a commonly used strategy, the probe was kept unchanged for the first five iterations. Since the illumination has a restricted NA easily estimated from the sum of all the forward ptychographic diffraction patterns, we also constrained the probe reconstruction in the Fourier space with a rectangular mask of 1.5 times the estimated NA size for each iteration (until 1000 iterations). The initial guess for the sample was a uniform slab. As the surfaces of the tungsten crystal foil were not parallel to the translational scan plane, the slab was tilted by 5.5° around the **x** axis and by −2° around the **y** axis following a right-handed convention (angles were estimated from some preliminary reconstructions). With the angular tilts applied, during the reconstruction, the slab thickness was enlarged in three successive steps: the thickness was set to 0.5 μm for the first 100 iterations, then increased to 0.7 μm for another 100 iterations, and finally increased to 0.9 μm for a third set of 100 iterations. This strategy helps with the early convergence of an accurate probe. An increasing thickness also accounts for any wrong estimation of the angular tilts and the irregular (larger) thickness of the crystal at the bottom part along **y**. Eventually, this also ends-up with a support too large in many places, which is detrimental in the end. This problem was solved with a final support refinement, similar to shrink-wrap[70]: from 300 to 500 iterations, the voxels in the object whose amplitude is lower than 15% of the maximum amplitude of the object reconstruction were set to zero every five iterations. After 500 iterations, the support function is fixed to the end of the reconstruction. Finally, a last step was performed, with 500 updates of the maximum likelihood

algorithm with Gaussian noise model[34]. The whole inversion is performed on a NVDIA high power computer (4 GPUs) and requires about 24 h to run.

For the reconstruction using a known, fixed probe, we followed the same procedures as above, except that the probe was not updated, and the angular up-sampling was not introduced.

**Extraction of strain and tilts.** The strain and tilts can be extracted from the 3D phase profile. They are derived as[28]:

$$\begin{cases} \epsilon_{zz}(\mathbf{r}) = |\mathbf{Q}_{220}|^{-1} \times \partial\phi_{220}(\mathbf{r})/\partial r_z \\ \omega_x(r) = \arcsin[|\mathbf{Q}_{220}|^{-1} \times \partial\phi_{220}(\mathbf{r})/\partial r_y] \\ \omega_y(r) = \arcsin[|\mathbf{Q}_{220}|^{-1} \times \partial\phi_{220}(\mathbf{r})/\partial r_x] \end{cases} \quad (14)$$

where $\epsilon_{zz}$ is the strain along the **z** axis, which is coincident with $\mathbf{Q}_{220}$, $\omega_x$ is the lattice rotation about **x** axis and $\omega_y$ the lattice rotation about **y** axis. As seen in Fig. 3, the extracted phase values are wrapped between $[-\pi, \pi]$. This wrapping would cause discontinuities when calculating the phase derivatives $\partial\phi_{220}(\mathbf{r})/\partial r$. To avoid this, the extracted phase needs to be unwrapped. However, phase unwrapping is problematic with the presence of dislocation defects due to their characteristic phase vortices[27,43]. Hence, the phase derivative was calculated using[71]

$$\partial\phi_{220}(\mathbf{r})/\partial r_* = \arg\{\exp[i\phi_{220}(\mathbf{r})] \times \exp[-i\phi_{220}(\mathbf{r} - r_*)]\}/r_*, \quad (15)$$

which does not require the phase to be unwrapped.

## Data availability

All Bragg ptychography data generated in this study have been deposited in the Zenodo database under accession code https://doi.org/10.5281/zenodo.5506676.

## Code availability

The Matlab routines, used for the Bragg ptychography approach presented here are available at https://doi.org/10.5281/zenodo.5506676.

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

## Acknowledgements

The BP-CDI experiment was performed on beamline ID01 at the European Synchrotron Radiation Facility (ESRF), Grenoble, France. This work received funding from the European Research Council (European Union's Horizon H2020 research and innovation program grant agreements No 724881 and No 714697, and under the Marie Skłodowska-Curie Actions grant agreement No. 884104 PSI-FELLOW-III-3i). The authors acknowledge use of characterisation facilities within the David Cockayne Centre for Electron Microscopy, Department of Materials, University of Oxford. Micro-beam Laue Diffraction experiments used the Advanced Photon Source, a US Department of Energy (DOE) Office of Science User Facility operated for the DOE Office of Science by Argonne National Laboratory under Contract No. DE-AC02-06CH11357. Ion implantations were performed at the Ion Beam Centre at the University of Surrey as part of United Kingdom Engineering and Physical Sciences Research Council grant EP/H018921/1. We are grateful to Hongbing Yu for help with sample preparation and Edoardo Zatterin for help during experiments. Vincent Favre-Nicolin is warmly acknowledged for his help during experiments and for fruitful discussions.

## Author contributions

V.C. designed the project. P.L. developed the inversion strategy, wrote the code, inverted and analysed the data with the help of M.A. F.H. and N.P. designed and prepared the samples and the synchrotron-based experiments. F.H., N.P., P.L., V.C., and S.L. were involved in the coherent diffraction experiment at ESRF. Micro-Laue and EBSD characterisation were performed by F.H. and N.P. V.C. and P.L. wrote the paper with the help of all co-authors.

## Competing interests

The authors declare no competing interests.
