## [Peer Review File · Nature Communications]

REVIEWER COMMENTS

Reviewer #1 (Remarks to the Author):

In the present study, the authors have developed an improved 3D Bragg ptychography, to examine the 3D density, phase, and displacement, strain, and rotation fields within materials. There, the field of view was extended and the data quality was improved, in comparison with the former similar technique. Also, the authors' method is better than HR-EBSD and X-ray micro-beam Laue diffraction, for the evaluation of strain field. The presented technique can be applied to nuclear-fusion materials (Ref. [56]). I sincerely admire the authors' great efforts for this elaborate work.

However, I am afraid that the presented technique is an incremental improvement, and I think that the results of its application (strain field due to "invisible defects" produced by He irradiation and dislocations in a W-Re alloy as a nuclear-fusion material) are not very impactful even in the strongly related field such as radiation damage or nuclear materials. Especially, it is not very clear how the obtained results can be attributed only to "invisible" defects, which is one of the authors' important claims. Therefore, I am sorry that in my current opinion the present paper looks more suitable to more specialized journals such as those for material analyses or nuclear technologies, rather than Nature Communications, at least in its present form.

I would like to suggest the points to be clarified for the improvement of the manuscript.

(1) How significant the progress of the authors' method is.

According to Fig. 3, the field of view (c, d) is actually extended in comparison with the former method (f, g). However, its extent looks only less than double.

In terms of data quality, the dislocations which are not visualized by the former method became visible by the authors' method. However, dislocations have already been visualized in another method shown in Ref. [43] authored by some authors of the present study.

(2) How the obtained strain field is connected to the defect concentration, and size distribution, etc.

If I correctly understand, the authors tried to connect the obtained strain field to the spatial distribution of defect concentration, and they concluded that the defect distribution was homogeneous. However, at the given resolution ($37 \times 40 \times 39 \text{ nm}^3$) which is much wider than the scale of individual defects, it must not be straightforward to extract the spatial distribution of the defect concentration under no information of the spatial distribution of the defect-size distribution. At least, the authors should consider that the initial size distribution of the defects produced by collision cascade must be dependent on the He incident energy. In the present study, the incident energy ranges from 0.05 to 1.8 MeV.

(3) How the presence of "visible" defects and their effects on the obtained strain are excluded. Even though the population of the bigger "visible" defects is low, the effect of individual bigger defects on the strain field must be more significant than that of individual smaller "invisible" defects (e.g. Ref. [56]).

The visible defects can be formed even for the examined He irradiation conditions. Firstly, the collision cascade can produce "visible" defects although their population is comparatively low (e.g. A. Sand et al. Euro Phys. Lett. 115(2016)36001.; the detailed defect-cluster size distribution must be changed by the difference of the incident-particle elements and energies.). Secondly, the elastic interaction among cascade-produced defects leads to the coalescence among defects (e.g. D. Mason et al. J. Phys. Cond. Matt. 26 (2014) 375701.), and it can yield bigger defects. The irradiation temperature (573 K) is not very low, which can enhance the motion of defects and the interaction among defects.

I think that the possibility of the presence of "visible" defects can be simply excluded by TEM.

(4) How homogeneous the strain field in Fig. 4 is.

For consideration of the grain-boundary effect on the defect distribution, at least the strain distribution across the grain boundary should be provided in the form as Fig. 4(i). According to Fig.

3(b), to my eyes, it does not look homogeneous but it looks lower at the region near the grain boundary.

Reviewer #2 (Remarks to the Author):

In this manuscript, in order to explore helium ion irradiation damage, the x-ray crystalline microscopy approach has been applied to image the lattice strains and tilts at the nano-scale 3D resolution. The results uncovered that the defects are likely isotropic in orientation and homogeneously distributed. In addition, no defect-denuded region was captured in the vicinity of the grain boundary. Before the publication, the authors should address the following comments from the reviewer.

1. During helium ion irradiation process, even if the accumulated helium concentration doesn't reach the critical value to form TEM detected bubbles, various defects will be generated, especially the vacancy clusters with different dimension. Is that possible for the proposed x-ray technique to differentiate the type of defects? Also, can it tell the difference between vacancy-type vs. interstitial-type defects?
2. From the SRIM (in the ref.) calculation, the distribution of damage and helium concentration changes as a function of the irradiation depth. However, from the calculated plots, it seems the strain distribution is uniform. Why?
3. In unirradiated region, can the authors tell the strain difference between grain boundary region and the region inside the grain? The phenomenon of defect denuded region at GBs is very common and has been verified from both experimental observation and modeling. This is correlated with the intrinsic properties of GBs to absorb radiation induced defects. What's the main scientific reason leading to the observed non defect-denuded region?
4. Here only ϵ_{zz} , ω_x and ω_y haven been calculated. Is there any possibility to get the strain value for the other components?

Reviewer #3 (Remarks to the Author):

Li et al., "Revealing 'invisible' defects in 1 implanted material with 3D Bragg ptychography"

This work reports on Bragg ptychography measurements of He implanted tungsten samples. The authors performed a complicated experiment and detailed analysis of the large amount of data collected in this experiment and report on strain fields observed in the tungsten sample in different regions of this sample: implanted and not implanted.

I have few major points that should be clarified before making any judgement on publication of this manuscript in Nature Communication journal.

1. In the title of the paper as well as in Abstract and Introduction the authors are talking about "...revealing 'invisible' defects in implanted material...". I was quite fascinated by this title and was hoping to see images of these otherwise "invisible" defects, but to my great disappointment, I haven't seen any images like that. This is very important point! I think, the authors should just change the title and rewrite the Introduction of the manuscript to state what is really observed in the manuscript! Otherwise, they should convince the reader that they really see "invisible" defects!

2. Other point is that authors present a new way of reconstructing the probe of the illumination together with the reconstruction of the sample. That is a good achievement and, from my point, deserves it's separate publication in a specialized journal. However, I do not understand in fact how probe is reconstructed in such featureless sample. Typically to reconstruct the probe you need a strong scattering and high contrast sample as Siemens star. In this work crystalline electron density is practically constant across the sample, so no strong features that can be observed, that is all putting a question on reliability of reconstructed probe!

3. The authors developed the algorithm that allows to reconstruct the probe together with the sample structure. Approach is based on the use of kinematical equation (1) of the main text. Thus reconstructed probe has the size of $5 \times 6 \mu\text{m}^2$. If we estimate the speckle size of such a probe according to known equation $\lambda L^2/D$, where λ is the wavelength, L is the sample-detector distance, and D is the area it will give us for a speckle size $43 \times 36 \mu\text{m}^2$ that is definitely less than the pixel size of detector that is for

Maxipix is $55 \times 55 \mu\text{m}^2$. It means that probe is not sufficiently sampled to this size! The authors should explain how they go around this important point in their probe reconstruction.

4. To my understanding ptychography scans were performed at different Bragg angular positions. For each angle sample was returned to its initial position and ptychography scans were repeated. For featureless sample as investigated in this work it is probably OK, but for more diverse sample it may be not a sufficient way of performing experiment, as step motors never come to the same position with the nanometer precision. Comment of the authors will be important to clarify this point.

5. The authors perform Bragg angular scans of their sample at different positions. I will be thankful if they will show the rocking curve in the region of pure Si wafer and in the region of the implanted part of the sample also indicating angles at which ptychography measurements were performed.

6. Related to that. I will be thankful if the authors will show reciprocal space maps collected in their experiment also both in Si wafer and implanted parts of the sample.

Some minor points:

1. In the 3-rd paragraph where the authors describe different ptychography measurements performed for different samples they also should cite the work: Dzhigaev et al., ACS Nano, **11**, 6605–6611 (2017), where ptychography approach was applied to imaging the strain field in Nanowires.

2. Line 130. The authors should write that experiment was performed at ESRF ID01 beamline.

3. On lines 131-132 the authors should provide correct focus size values at the sample (taking also into account Bragg angle).

4. Lines 137 and 140. Words "sufficiently small" and "sufficiently large distance" should be substituted by the numbers.

5. In Eq. (1) $\rho(\mathbf{r})$ is not specified after equation.

6. I think also that authors should be careful with using the term "electron density", but use instead rather "crystalline electron density". This is especially valid for the results presented for example in Figs. 3 and 4, where the bottom left part of the sample is "invisible" due to stacking fault. However electron density is not zero in this region as it is well seen in Fig. 1a.

7. Fig. 1. In the inset of Fig. 1a scale bar should be given. In panel Fig. 1b box of monochromator unit should be substituted by two crystals. Coordinate system p_1 , p_2 , p_3 , should be explained in Fig. caption.

8. Fig. 2. Line 852. In Fig. caption θ_1 is missing.

9. Fig. 3. It is not clear at which depth panels (c, d, f, g) are taken. It should be clearly indicated in Fig. caption. In panel(c, d) and (f, g) axes should be provided. In panels (d,g) the authors show the phase map from the reconstruction that has low significance due to wrapping of phase. It will be important if they will show the displacement field at least in some region of reconstructed part of the sample, then it will be easier to understand the flat strain fields shown in next Fig.

Resubmission of NCOMMS-20-29429 “Revealing nano-scale lattice distortions from ‘invisible’ defects in implanted material with 3D Bragg ptychography” by P. Li *et al.*

We are grateful to the referees for their careful review of our manuscript and their detailed comments regarding our work. We were delighted that all three referees commented positively on our work. In this revised version we have addressed the comments and questions they raised. In particular we emphasise (1) how our new Bragg ptychography approach allows going beyond the bottleneck imposed by the previous inversion scheme and (2) the interpretation of our findings with respect to defect analysis in ion-implanted materials. In addition to a revised text, the present manuscript includes four new figures focusing on the data quality, the displacement field, the agreement between the measured strains and those predicted by the simulations and the grain boundary region. We feel that we have fully addressed the points raised by the referees and hope that this revised version convincingly conveys the importance and impact of our results.

RESPONSE TO REVIEWER #1

We thank Reviewer #1 for the careful reading of the manuscript and we understand from his/her comments that we should better highlight the importance of the breakthrough we led with respect to crystalline microscopy methods and the relevance of our findings on establishing the presence of “invisible” defects in ion implanted materials. We reply below to all his/her comments and questions. The corresponding modifications in the text are highlighted in blue.

General comments: *“In the present study, the authors have developed an improved 3D Bragg ptychography, to examine the 3D density, phase, and displacement, strain, and rotation fields within materials. There, the field of view was extended and the data quality was improved, in comparison with the former similar technique. Also, the authors’ method is better than HR-EBSD and X-ray micro-beam Laue diffraction, for the evaluation of strain field. The presented technique can be applied to nuclear-fusion materials (Ref. [56]). I sincerely admire the authors’ great efforts for this elaborate work. However, I am afraid that the presented technique is an incremental improvement, and I think that the results of its application (strain field due to “invisible defects” produced by He irradiation and dislocations in a W-Re alloy as a nuclear-fusion material) are not very impactful even in the strongly related field such as radiation damage or nuclear materials. Especially, it is not very clear how the obtained results can be attributed only to “invisible” defects, which is one of the authors’ important claims. Therefore, I am sorry that in my current opinion the present paper looks more suitable to more specialized journals such as those for material analyses or nuclear technologies, rather than Nature Communications, at least in its present form. I would like to suggest the points to be clarified for the improvement of the manuscript.”*

We are grateful to Reviewer #1 for his/her positive comments on the development of the method and resulting data quality. As acknowledged by Reviewer #1, we are not aware of any other approach able at providing such a detailed evaluation of the strain and lattice plane tilts in an extended crystalline sample. Let us underline that this was made possible by the introduction of the simultaneous retrievals of the probe and sample scattering functions during the inversion scheme, a long-standing problem in Bragg ptychography. This problem is particularly challenging due to the lack of information contained in the 2D Bragg ptychography scan, this 2D information being exploited to retrieve two 3D functions. To our knowledge, our work pioneers this development. We are able to achieve this by combining a relevant assumption of the probe behaviour with the up-sampling of the data set, two technically challenging operations as a result of the non-orthogonal 3D description of the data-sets. Our approach was thoroughly developed on numerical tests. The quality of the retrieved images is foreseen to catch interest of an extended material science community.

In the present study, we do not only demonstrate this new method, but also exploit it to shed light on ion-implantation in materials. We are able to provide an unprecedented high quality 3D image of an ion-implanted crystal and of its neighbouring reference material. This led to an estimate of the concentration of the invisible defects, found to be randomly orientated and resulting in no prominent denuding near the grain boundary. These new findings are key insights essential to inform the design of next generation radiation-resistant materials. Due to the importance of our findings with respect to defect characterization in crystalline materials and the potential impact of the Bragg ptychography method in other material science related problems, we believe that our manuscript is of interest for a large readership. We hope this revised version better convey this idea.

We have modified the abstract to better highlight the importance of our work: “Our results lead to the conclusions that few-atom-large ‘invisible’ defects are likely isotropic in orientation and homogeneously distributed. No defect-denuded region is observed close to the grain boundary. These new findings open up exciting...”.

The introduction has been revised too. It now contains: “However, these predictions cannot be verified by existing experiments” and “Whilst we still cannot directly resolve ‘invisible defects’, we unambiguously probe them via the strain fields they cause, which extend over distances much larger than the defects themselves, thus capturing their presence and characteristics. As such, we can assess their behaviour in an unmatched manner.”

Comments #1: “How significant the progress of the authors’ method is.

According to Fig. 3, the field of view (c, d) is actually extended in comparison with the former method (f, g). However, its extent looks only less than double.

In terms of data quality, the dislocations, which are not visualized by the former method, became visible by the authors’ method. However, dislocations have already been visualized in another method shown in Ref. [43] authored by some authors of the present study.”

The comment on the field of view results from an unfortunate configuration of our crystal. From this comment, we realized we should have been more explicit to explain that the field-of-view increase is indeed close to a factor of 2. Fig. 3c and d shows only a sub-region of the total possible field of view, which is occupied by the grain for which the Bragg condition is satisfied. Up-sampling enables an increase of the field of view based on an increase in the maximum extent of the object primarily in the x direction, in the present case from 4.2 μm to 7.8 μm (i.e., a factor of 1.85). In the present system, the increased field of view largely covers the neighbouring grain, which has a different orientation and is therefore not visible. The maximum field of view enabled by the angular up-sampling approach is shown in S3d vs S3e as well as the selected sub-region, which is used in the subsequent main text figure. The main text contains the following statement when Fig. 3 is introduced “For sake of clarity, only the relevant part of the retrieved image is shown, using a mask applied to all presented maps (see Supplementary information S3 for the mask definition)”.

We have added the following text to the figure caption to make this explicitly clear:

“Note the strong reduction of the field of view (evidenced by the dashed rectangle) and the degradation of the image quality. A comparison of the total possible field of view is shown in supplementary Figure 3d, including regions of the possible field of view that are non-scattering due to the finite extent of the crystal.”

Regarding the second part of Comments #1, Reviewer #1 is correct that the visualisation of dislocations using (non-ptychographic) BCDI has previously been demonstrated. However, in these previous studies only sub-micron volumes could be probed. The present sample volume is about 2 orders of magnitude larger! Such extended samples cannot be probed by non-ptychographic BCDI. To clarify the significance of our work:

- Firstly, this is the first time Bragg ptychography (BP) has enabled the visualization of the 3D strain fields of multiple dislocations in an extended object. This offers the potential to investigate defects, defect networks and defects interaction in a larger volume, providing insight into the local environment. A key factor here is that sufficiently large sample sizes are key in order to avoid surface artefacts that appear when defect behaviour and structure change due to the elastic image fields associated with free surfaces. In TEM these “free surface artefacts” are a frequent problem [see e.g., Mason *et al.*, *J. Phys.: Condens. Matter* **26**, 375701 (2014) and H. Yu *et al.*, *Ultramicroscopy* **195**, 58 (2018)].
- Secondly, in relation to the visibility of dislocations within the current system. Here our study shows that standard strain mapping methods fail to detect the presence of the dislocations (see comparison between HR-EBSD, micro-beam Laue X-ray diffraction and BP in supplementary section 4) whereas BP clearly resolves these important defects.
- Thirdly, the BP method developed in this study affords, for the first time, both sufficient strain sensitivity and spatial resolution to directly probe the strain fields resulting from He implantation within a 3D volume. The effects of low-dose He implantation, whilst known to affect material properties [D.E.J. Armstrong *et al.*, *Appl. Phys.Lett.* **251901**, 1 (2013), F. Hofmann *et al.*, *Scientific Reports* **5** 16042 (2015)], remain invisible to TEM, which lacks sensitivity to detect the resulting few-atom-large defects [Z. Zhou *et al.*, *Philos. Mag.* **86**, 4851 (2007), D.E.J. Armstrong *et al.*, *Appl. Phys.Lett.* **251901**, 1 (2013)]. Some limited success probing these defects has been reported by positron annihilation spectroscopy [A. Debelle *et al.*, *Journal of Nuclear Materials* **362**, 181 (2007), A. Debelle *et al.*, *Journal of Nuclear Materials* **376**, 216 (2008), P. E. Lhuillier *et al.*, *Phys. Status Solidi C* **6**, 2329 (2009)] or indeed X-ray micro-diffraction to probe defects via the strain fields they cause [F. Hofmann *et al.*, *Acta Materialia* **89**, 352 (2015), I. De Broglie *et al.*, *Scripta Materialia* **107**, 96 (2015), S. Das *et al.*, *Materials & Design* **160**, 1226 (2018)]. However, these techniques lack spatial resolution and cannot resolve the fine detail of where helium-implantation-induced defects accumulate, whether they accumulate uniformly, or whether there is a reduced concentration of defects at sinks, such as grain boundaries (an effect that has been reported for larger defects such a nano-metre sized dislocation loops). BP measurements are transformational, as they provide both excellent strain and spatial resolution. In addition to the observation of a series of dislocations within the implanted and non implanted region, BP makes it possible to determine the following new findings: a) an estimate of the invisible defect concentration, b) that these invisible defects are of random orientation and c) that there is no prominent denuding of these defects near the grain boundary. These are key insights essential to inform the design of next generation radiation-resistant materials.

To better underline the novelty our approach provides regarding the dislocations pointed out in Comments #1, we have modified the introduction which now includes: “This approach allows us to extend the field view while improving the image quality, making it possible to directly compare implanted and non-implanted sample regions. This comparison highlights details of the crystalline structure, such as lattice damage from helium irradiation, several dislocations and sample preparation damage.”

Comments #2: “*How the obtained strain field is connected to the defect concentration, and size distribution, etc.*”

If I correctly understand, the authors tried to connect the obtained strain field to the spatial distribution of defect concentration, and they concluded that the defect distribution was homogeneous. However, at the given resolution (37×40×39 nm³), which is much wider than the scale of individual defects, it must not be straightforward to extract the spatial distribution of the defect concentration under no information of the spatial distribution of the defect-size distribution. At least, the authors should consider that the initial size distribution of the defects produced by collision cascade must be dependent on the He incident energy. In the present study, the incident energy ranges from 0.05 to 1.8 MeV.”

The defects observed within this study fall into two main classes (noting that defects generated by FIB damage are excluded from this consideration):

Dislocations: These are clearly discernible in the BP data by a characteristic phase vortex with a central pipe of missing intensity. Here we confirm the type by comparison with an elastic dislocation model (see S8).

He-induced defects: Previous theoretical and experimental work has shown that He-implantation of W leads to predominantly low primary knock-on atom (PKA) energies. This is evident from the plot below, extracted from [F. Hofmann *et al.*, *Acta Materialia* **89**, 352 (2015)], which shows the recoils per He ion per eV plotted as a function of the PKA energy for different depths in the sample. The He energies considered in this plot are the same as those used in the present sample.

This plot shows that He-ion implantation, even at 1.8 MeV, results predominantly in low PKA energies as reproduced from [F. Hofmann *et al.*, *Acta Materialia* **89**, 352 (2015)]. For comparison typical self-ion damage PKA energies that have been considered in previous studies are on the order of 150 keV to 200 keV [A. Sand *et al.*, *Euro Phys. Lett.* **115**, 36001 (2016)]. This means that in the case of He implantation the damage microstructure is dominated by Frenkel pair generation, and no larger defects, such as those reported for self-ion implantation are expected. A further discussion of the defects formed by He-implantation is provided in response to point 1.3.

[Redacted]

The volumetric lattice strain due to small lattice defects can be computed by over the sum of defect relaxation volumes [S. L. Dudarev *et al.*, *Nuclear Fusion* **58**, 126002 (2018)]. Previous DFT and MD calculations have shown that, for small clusters of vacancies and interstitials (less than ~50 defects), the relaxation volume per defect changes little with the number of defects in the cluster [F. Hofmann *et al.*,

Acta Materialia **89**, 352 (2015)]. As such, in the present case, given that we know all He-induced defects fall into this regime, the precise size distribution of defects does not need to be considered. Rather an

“equivalent” Frenkel pair density can be determined, i.e. the density of isolated vacancies (V) and self-interstitial atoms (SIA) pairs that lead to the lattice swelling. For further details of the calculation of Frenkel pair density, we would kindly direct the reviewer to the response to Reviewer #2, Comments #2. As discussed below both the main text has been revised and details of the calculation are provided in the text accompanying supplementary Fig. S9.

Comments #3: *“How the presence of “visible” defects and their effects on the obtained strain are excluded.*

Even though the population of the bigger “visible” defects is low, the effect of individual bigger defects on the strain field must be more significant than that of individual smaller “invisible” defects (e.g. Ref. [56]).

*The visible defects can be formed even for the examined He irradiation conditions. Firstly, the collision cascade can produce “visible” defects although their population is comparatively low (e.g., A. Sand et al. Euro Phys. Lett. **115**, 36001 (2016); the detailed defect-cluster size distribution must be changed by the difference of the incident-particle elements and energies). Secondly, the elastic interaction among cascade-produced defects leads to the coalescence among defects (e.g. D. Mason et al. J. Phys. Cond. Matt. **26** (2014) 375701), and it can yield bigger defects. The irradiation temperature (573 K) is not very low, which can enhance the motion of defects and the interaction among defects.*

I think that the possibility of the presence of “visible” defects can be simply excluded by TEM.”

The paper by Sand *et al.* [A. Sand et al. Euro Phys. Lett. **115** 36001 (2016)] considers collision cascades with 140 to 200 keV PKA energy. These produce a range of defects - captured by the now well-known power law behaviour. Our previous calculations (see [F. Hofmann *et al.*, Acta Materialia **89**, 352 (2015)] and plot shown in response to 1.2) showed that for He irradiation PKA energies are much lower, predominantly less than ~500 eV. As such, for He ion implantation, Frenkel pair generation is the predominant damage formation mechanism.

Under the present implantation conditions, vacancies in tungsten are essentially immobile, while SIAs are mobile even at cryogenic temperatures [K.D. Rasch *et al.*, Philos. Mag. A **41**, 91 (1980), A. Debelle *et al.*, Journal of Nuclear Materials **376**, 216 (2008)]. He has a strong affinity to vacancies, to which it binds. Since a population of vacancies are now occupied by He, these cannot readily recombine with SIAs. Instead SIAs and He-filled vacancies form stable defect structures [L. Sandoval *et al.*, Phys. Rev. Lett. **114**, 105502 (2015)]. These are too small to be imaged directly by TEM, as confirmed by TEM observations of tungsten implanted with He under similar conditions in [D.E.J. Armstrong *et al.*, Appl. Phys.Lett. **251901**, 1 (2013)].

We do agree that quantification of defects in the immediate vicinity of the larger dislocations is challenging, given that the dislocation strain fields are significant up to ~200 nm from the dislocation core. As such this region has not been included in our analysis.

This is reflected in the text in the introduction: *“Whilst we still cannot image ‘invisible defects’, we unambiguously probe them via the strain fields they cause, which extend over distances much larger than the defects themselves thus capturing their presence and characteristics. As such, we can assess their behaviour in an unmatched manner.”*

and further down in the Discussion section:

“Noticeably, this strain can be distinguished from that of larger dislocations so that strain from dislocations can be excluded from the consideration of strain due to invisible defects.”

Comments #4: “How homogeneous the strain field in Fig. 4 is.

For consideration of the grain-boundary effect on the defect distribution, at least the strain distribution across the grain boundary should be provided in the form as Fig. 4(i). According to Fig. 3(b), to my eyes, it does not look homogeneous but it looks lower at the region near the grain boundary.”

We have added the requested figure, which appears in the new Figure 5 (as 5d). While the strain decays slightly when approaching the grain boundary, there is no indication that this corresponds to the possible denuded region (i.e., we observe a weak decay and an extended length scale). We have thought a lot about this point, as we were expecting to see a clear denuded layer, where the strain plateaus and is similar to the un-implanted region (i.e., close to 0) before rising to the implanted value. Since the strain in the denuded layer would be similar to that in the unimplanted material, we would not have missed this region in our reciprocal space sampling, *i.e.* the signal from the denuded layer would have fallen into the solid angle spanned by the diffraction detector. However, given the spatial resolution of our retrieved image, we cannot unambiguously conclude on the absence of a denuded region. If any, the denuded region is neither thick nor strongly unstrained.

Modification to the text: The following text has been added in the discussion part: “Within the limit of the achieved spatial resolution, we can safely state that the denuded region, if it exists, is neither strongly unstrained nor spatially extended.”

A new strain profile has been extracted as requested and added in the new Figure 5 as Figure 5d. The new figure caption reads:

“Figure 5: Extracting displacement fields and strain profiles. (a) *Left* zoomed-in 3D iso-surface density plots of the dislocation #1 highlighted by a white rectangle in the implanted layer, shown in Fig. 4b, d and f, *middle* 2D cross-section map of the ϕ_{220} reconstructed phase and *right* estimated phase variation resulting from simulation (see Supplementary information S8). (b) Zoomed-in view of the ϵ_{zz} strain map extracted from the implanted region in the vicinity of the grain boundary (as shown in Fig. 5b). (c, d, e) One-dimensional cross sections of the average ϵ_{zz} strain map, across the film thickness, perpendicular to the grain boundary within the implanted region and along the implantation direction, respectively, as indicated in Fig. 4b. In (e) the grey area corresponds to a region where ϵ_{zz} is slightly degraded due to a bit of parasitic aliasing along the *y* direction. (f) Calculated strain profile along the implantation direction, assuming the damage microstructure consists of Frenkel pairs with vacancy filled by 1 He atom (SIA + VHe) or as Frenkel pairs alone (SIA + V). See Supplementary information S9 for details. All scale bars and angular colour scales in radian are indicated on the plots.”

Furthermore, a new figure dedicated to the strain profile at the grain boundary has been added (as Supplementary Information S9).

RESPONSE TO REVIEWER #2

We thank Reviewer #2 for the careful reading of the manuscript and his/her positive opinion on our work. We have taken into account all remarks from Reviewer #2 and reply below to all comments and questions raised by him/her. The corresponding modifications in the text are highlighted in green.

General comments: “*In this manuscript, in order to explore helium ion irradiation damage, the x-ray crystalline microscopy approach has been applied to image the lattice strains and tilts at the nano-scale 3D resolution. The results uncovered that the defects are likely isotropic in orientation and homogeneously distributed. In addition, no defect-denuded region was captured in the vicinity of the*

grain boundary. Before the publication, the authors should address the following comments from the reviewer.”

Thanks for positive judgment of our work. The comments are addressed below.

Comments #1: “During helium ion irradiation process, even if the accumulated helium concentration doesn’t reach the critical value to form TEM detected bubbles, various defects will be generated, especially the vacancy clusters with different dimensions. Is that possible for the proposed x-ray technique to differentiate the type of defects? Also, can it tell the difference between vacancy-type vs. interstitial-type defects?”

We agree that under the present conditions no He bubbles are expected to form. TEM observations of tungsten irradiated with He under similar conditions, but to a substantially higher dose, though still below the limit for bubble formation, showed no visible defects [D.E.J. Armstrong *et al.*, Appl. Phys. Lett. **251901**, 1 (2013)]. This agrees with theoretical calculations that show that in the present sample Frenkel pair generation will probably be the dominant damage mechanism [F. Hofmann *et al.*, Acta Materialia **89**, 352 (2015)]. He then occupies the vacancies (V), preventing their recombination with self interstitial atoms (SIAs). Instead He-filled vacancies and SIAs form stable complexes [L. Sandoval *et al.*, Phys. Rev. Lett. **114**, 105502 (2015)].

The volumetric lattice strain arising from atomic scale defects can be captured by summing the defect relaxation volumes over the defect population [S. L. Dudarev *et al.*, Nuclear Fusion **58**, 126002 (2018)]. SIAs have a large and positive relaxation volume (1.68), so lead to a lattice expansion. Vacancies, on the other hand, have a smaller, negative relaxation volume (-0.38) leading to a lattice contraction [F. Hofmann *et al.*, Acta Materialia **89**, 352 (2015)]. Thus, since we observe a lattice swelling, there must be some SIAs retained in the lattice. However, given that SIAs are highly mobile while vacancies are sessile in the present experiments, there must be at least one vacancy per SIA in the lattice. Thus, we can obtain a lower bound estimate of defect number density (700 appm), by considering the density of Frenkel pairs required to obtain the observed lattice strain.

An interesting question concerns clustering of defects. For small defects below the TEM visibility, previous theoretical work has shown that the relaxation volume per defect is not a strong function of clustering [F. Hofmann *et al.*, Acta Materialia **89**, 352 (2015)]. This means it is justifiable to consider the strain effect due to He-induced defects in terms of an equivalent Frenkel pair density, where each V and SIA is considered in isolation. This is the approach taken here.

In order to answer this comment and Comments #2 below, the manuscript has been modified and now reads (in the Discussion section):

“This lattice expansion can be understood in terms of Frenkel defects where He occupies the vacancy, preventing recombination. Atomistic simulations have suggested that the He-occupied vacancy and self-interstitial atom (SIA) form a stable, bound configuration. The net positive lattice swelling arises as self-interstitials have large positive relaxation volume (Ω_r (SIA) = 1.68), while vacancies, with or without He, have smaller negative relaxation volume (Ω_r (V) = -0.37, Ω_r (V+He) = -0.24). The lower bound Frenkel pair density, required to produce the measured strain, can thus be estimated as ~700 appm (see Fig. S9 for details). Comparing to the damage predicted by binary collision simulations (Fig. S9), this suggests only ~4% of the induced irradiation damage is retained. Using this retention efficiency, the lattice strain profile anticipated from the depth-variation of damage in binary collision simulations can be predicted (Fig. 5f). The agreement with the measured strain profile is quite remarkable, indicating that, at least on the scale of our measurements, there is no appreciable defect migration.”

Comments #2: “From the SRIM (in the ref.) calculation, the distribution of damage and helium concentration changes as a function of the irradiation depth. However, from the calculated plots, it seems the strain distribution is uniform. Why?”

The damage distribution in the implanted layer predicted by SRIM is actually quite uniform, as seen in the figure below extracted from Das et al. [S. Das *et al.*, *Materials & Design* **160**, 1226 (2018)] for the same implantation profile. A similar figure, showing the depth-variation of damage and injected helium concentration has now been included in the Supplementary Information as Figure S8.

[Redacted]

Figure: Profile of injected helium ion concentration (solid blue curve) as calculated by SRIM and implantation-induced displacement damage (dotted grey curve) as a function of depth in the sample.
Extracted from Das et al. (2018).

While the damage is quite uniform, the injected He concentration, predicted by SRIM, shows a number of distinct peaks. Since the damage profile is quite uniform, a reasonably uniform distribution of the associated lattice strains is expected. This is because vacancies are effectively immobile at the present implantation conditions [K.D. Rasch *et al.*, *Philos. Mag. A* **41**, 91 (1980), A. Debelle *et al.*, *Journal of Nuclear Materials* **376**, 216 (2008)]. He in interstitial form, on the other hand, is rather mobile by comparison. As such, one would expect He to move around the lattice until it finds a vacancy to get bound to. Previous theoretical work indicated that the resulting complexes, consisting of He-filled V with SIAs bound to them in close vicinity are rather stable [L. Sandoval *et al.*, *Phys. Rev. Lett.* **114**, 105502 (2015)].

It is interesting to compare the experimentally measured strain in the sample as a function of position along the implantation direction, and the strain anticipated based on SRIM calculations. To this aim, the damage microstructure was approximated as either consisting of Frenkel pairs where the vacancy is filled with 1 He atom (equation 1), or as Frenkel pairs alone (equation 2).

$$\epsilon_{zz} = 1/3 * (\Omega_r(\text{SIA}) + \Omega_r(\text{V+He})) * \text{damage} * \text{recombination} \quad (1)$$

$$\epsilon_{zz} = 1/3 * (\Omega_r(\text{SIA}) + \Omega_r(\text{V})) * \text{damage} * \text{recombination} \quad (2)$$

‘damage’ here refers to the damage at a particular depth in the sample. The only free parameter is ‘recombination’, a fixed constant that describes the proportion of the SRIM-predicted damage that is retained in the material. It is worth noting that this expression or strain is different to that provided in [F. Hofmann *et al.*, *Acta Materialia* **89**, 352 (2015)]. The reason is that in the present case the sample preparation has removed the lateral constraints incorporated in the expression in Hofmann *et al.*, (2015).

The plot shown below (as Fig. 5f and Supplementary Information S8) was obtained for a constant recombination value of 3.8%. That is of the ~66 Frenkel pairs generated per injected He, as predicted by SRIM, only ~2.5 Frenkel pairs per He are retained. As such we have a He : Frenkel pair ratio of 1:2.5, which justifies the two limits considered in equations (1) and (2). The plot below shows these two limits, using the same recombination value of 3.8%. The agreement with the experimentally measured strain profile is remarkably good between depths of 0.5 and 3.5 microns.

Modification to the text: The figure shown above has been added as Fig. 5f and Supplementary Information S8, together with the details of the SRIM calculations.

The text has been modified (see text modification in our answer to Comments #1 above).

Figure Caption

He implantation profile (green) and resulting strain profiles assuming that the damage microstructure is either consisting of Frenkel pairs where the vacancy is filled with 1 He atom (eq. 1, red) or as Frenkel pairs alone (eq. 2, blue). Those calculated strain profiles are compared to the experimental strain profile, extracted along the implantation direction (black). The overall amplitude, shape and finer details are in good agreement with the models. Note that the grey area corresponds to a region where a bit of aliasing is degrading locally the retrieved strain profile. The arrows correspond to the relevant axis for each plot. Further details on the calculation are provided above.

Comments #3 “*In un-irradiated region, can the authors tell the strain difference between grain boundary region and the region inside the grain? The phenomenon of defect denuded region at GBs is very common and has been verified from both experimental observation and modeling. This is correlated with the intrinsic properties of GBs to absorb radiation induced defects. What’s the main scientific reason leading to the observed non defect-denuded region?*”

This is a very interesting question. We have thought a lot about this point, as we were expecting to see a clear denuded layer, where the strain is similar to the un-implanted region. Since the strain in the denuded layer would be similar to that in the unimplanted material, we would not have missed this region in our reciprocal space sampling, *i.e.* the signal from the denuded layer would have fallen into the solid angle spanned by the diffraction detector.

Our current hypothesis for the lack of denuding is as follows: Defect denuded zones near grain boundaries have been observed for larger defects, such as dislocation loops formed by collision cascades [Z. Zhang *et al.*, *J. Nucl. Mater.* **480**, 207 (2016)]. The elastic lattice strain fields, and hence the forces driving defects to sinks, depend on the defect size [D. Mason *et al.*, *J. Phys. Cond. Matt.* **26**, 375701 (2014)]. Larger defects experience a larger driving force to sinks and also have longer range strain fields. This would then suggest that more significant denuding might be expected for larger defects, whilst any denuded zone for small defects would be shallower. An additional factor in the present experiments is that He-filled vacancies have a high migration energy [D. Perez *et al.*, *Scientific Reports* **7**, 2522 (2017)],

meaning that once created they do not readily move, restricting their migration to sinks. This is our current working hypothesis, though further theoretical and experimental work will be required to confirm or disprove this.

In order to clarify this point, we have modified the text (Discussion section), which now reads: “Our results suggest that for small defects and defect clusters, such as those created by helium ion-irradiation, this is not the case. Within the limit of the achieved spatial resolution, we can safely state that the denuded region, if it exists, is neither strongly unstrained nor spatially extended. One possibility is that defect size affects denuded zone width, whereby small defects, which experience smaller elastic driving forces to sinks, lead to shallower denuded zones. Additionally, the high migration energy of He-filled vacancies may restrict the growth of denuded zones. Either way, our observations suggest that the potential of grain size reduction as a means of reducing irradiation defect accumulation is limited.”

We have also added a dedicated figure as Supplementary Information S9, to further describe the strain profile at the grain boundary in the crystal implanted and non-implanted regions

Comments #4: “Here only ϵ_{zz} , ω_x and ω_y have been calculated. Is there any possibility to get the strain value for the other components?”

The measurement of strain from Bragg coherent diffractive imaging provides the means to interrogate the lattice displacement perpendicular to the Bragg vector. From the projection of lattice displacement, some components of strain and lattice rotation can be calculated. In the present case, a single specular 220 reflection has been measured which provides the ϵ_{zz} , ω_x and ω_y components. In principal, data from more reflections could be combined to probe the all of the remaining components and would require the measurement of a minimum of 3 non-perpendicular reflections e.g. [M. Newton *et al.*, Nature Materials 9 120 (2010), F. Hofmann *et al.*, Scientific Reports 7, 45993 (2017)]. While this is in principle possible, it remains technically challenging with the state-of-the art synchrotron experimental set-ups is yet to be demonstrated.

In order to add this complementary information in the revised version, we have included (end of the Results section): “Note that the full access to the strain and tilts components would require the investigation of at least three non-orthogonal Bragg reflections, a technically challenging though conceivable experiment.”

RESPONSE TO REVIEWER #3

We thank Reviewer #3 for the careful reading of the manuscript and we understand from his/her comments that some parts in the original version of the manuscript needed to be improved to gain clarity. We agree with this statement and we revised the manuscript accordingly. We reply below to all comments and questions raised by the Reviewer. The corresponding modifications in the text are highlighted in orange.

General comments: “This work reports on Bragg ptychography measurements of He implanted tungsten samples. The authors performed a complicated experiment and detailed analysis of the large amount of data collected in this experiment and report on strain fields observed in the tungsten sample in different regions of this sample: implanted and not implanted.

I have few major points that should be clarified before making any judgment on publication of this manuscript in Nature Communication journal.”

From these general comments, we understand that Reviewer #3 acknowledges the difficulty of the experiment and the efforts we made to extract structural information and analyze them in the context

of ion implantation. We would like to take the opportunity to underline this was made possible because we pushed the Bragg ptychography approach beyond its former limit, by breaking a long-standing statement *i. e.*, the impossibility to retrieve simultaneously (and with high fidelity) both 3D sample and probe in this microscopy modality. This breakthrough was achieved by using the invariance of the incident beam while it propagates through the sample, a fully relevant assumption given the sample thickness and numerical aperture of the x-ray focusing devices. This new approach allows us to tremendously improve the sample image quality not only because we can now disentangle accurately sample and probe signals but also because we can account for the signal contribution arising from the extended (however low intensity) tails of the beam, a contribution thought to be negligible so far. This provided us with a non-anticipated gain in the accessible field-of-view. We believe it broadens the scope of Bragg ptychography.

To better highlight this achievement, we have added in the introduction: “Moreover, BP requires the 3D probe to be known prior to the sample electron density reconstruction, **a strong limit, which has not been overcome so far.**”

The conclusion has been updated along this line, too. It now reads: ‘**Our new inversion scheme provides an improved sample image quality, not only because the probe and sample can now be accurately disentangled but also because the signals arising from regions far the probe central lobe are now accounted for.**’

Comments #1: *“In the title of the paper as well as in Abstract and Introduction the authors are talking about “...revealing “invisible” defects in implanted material...”. I was quite fascinated by this title and was hoping to see images of these otherwise “invisible” defects, but to my great disappointment, I haven’t seen any images like that. This is very important point! I think, the authors should just change the title and rewrite the Introduction of the manuscript to state what is really observed in the manuscript! Otherwise, they should convince the reader that they really see “invisible” defects!*

We understand that our former title was unfortunately creating some frustration. To fully answer this comment, let us note that substantial work has been previously dedicated to exploring the high helium concentration regime (above ~6000 appm), where the formation of helium bubbles dominates the damage microstructure. A rather more complex scenario is however associated with lower helium concentrations that are expected to occur inside future fusion reactor components due to He generated by transmutation [M. R. Gilbert *et al.*, Nuclear Fusion 52, 083019 (2012)]. Previous studies of He doses up to 3000 appm, implanted at room temperature showed no visible defects in TEM [D.E.J. Armstrong *et al.*, Appl. Phys. Lett. 102, 251901 (2013)]. This is consistent with theoretical predictions that suggest defects in this case will consist of He-filled Frenkel defects, with only little clustering (*i.e.*, small defects). Because TEM is not sufficiently sensitive to probe these small defects [Z. Zhou *et al.*, Philos. Mag. 86, 4851 (2006)], characterizing this damage, which still leads to remarkable changes in mechanical and physical properties, has been a major challenge.

In this context, Bragg ptychography is transformational, since the excellent strain sensitivity and the 10s of nm 3D spatial resolution we are able to directly probe these otherwise “invisible” defects via the strain fields they cause, which extend over distances much larger than the defects themselves. As such, whilst we do not provide images of individual defects, we are uniquely able to probe their spatial distribution at the nano-scale. Furthermore we can assess whether these defects show preferential orientation - they don’t - and whether they are depleted near sinks, such as grain boundaries - they are not.

According to this comment, we have decided to modify the title, which now is “Revealing nano-scale lattice distortions from ‘invisible’ defects in implanted material with 3D Bragg ptychography”.

The introduction has been modified as well, to better highlight the novelty of our findings. It now reads (end of introduction):

“Whilst we still cannot directly resolve ‘invisible defects’, we unambiguously probe them via the strain fields they cause, which extend over distances much larger than the defects themselves, thus capturing their presence and characteristics. As such, we can assess their behaviour in an unmatched manner.”

Comments #2: *“Other point is that authors present a new way of reconstructing the probe of the illumination together with the reconstruction of the sample. That is a good achievement and, from my point, deserves its separate publication in a specialized journal. However, I do not understand in fact how probe is reconstructed in such featureless sample. Typically to reconstruct the probe you need a strong scattering and high contrast sample as Siemens star. In this work crystalline electron density is practically constant across the sample, so no strong features that can be observed, that is all putting a question on reliability of reconstructed probe!”*

We thank the reviewer for acknowledging the importance of this achievement. However his/her comments show that our text can be improved to better convey the relevance of our results. Regarding the featureless behavior of our sample, we would like to recall that the shape of the diffraction pattern, in 3D, is dependent on two factors: the (crystalline) electron density and the phase structure (associated to the lattice distortions in the Bragg geometry). Although the electron density of a crystal is often very homogenous - as highlighted by the referee -, on the contrary, the phase structure can be very complex, as a direct witness of the crystalline distortions. In the case presented in this manuscript, the phase structure was indeed very complex (see Fig. 3d). This was already indicated by the fact that the diffraction patterns are strongly different from what is expected for a homogeneously strained crystal (*i. e.*, the Fourier transform of the probe) and moreover, the diffraction patterns are different from one position to another (see the diffraction patterns in figure 1c of the manuscript and also in the reply Comments #6, below). On top of that, the crystal, which we imaged, presents two grain-boundaries (see figure 3), which create strong contrast in the electron density and also facilitate the reconstruction of probe function.

Another argument, which shows the robustness of our inversion procedure, was described in Fig. S1 (Supplementary Materials): we demonstrated that the probe reconstruction is successful even when the probe initial guesses is much different from the true solution.

Finally, in forward ptychography geometry, a lot of imaging has been performed on pure phase structures such as biological samples (see *e.g.*, [K. Giewekemeyer *et al.*, PNAS 107, 529 (2010), J. Deng *et al.*, PNAS 112, 2314 (2015)]) for which both probe and sample were retrieved simultaneously and with high fidelity.

In order to better convey the relevance of our finding with respect to the nature of the sample, we have modified the text, which now reads, in the introduction:

“Although single-crystalline samples present a constant electron density, the phase associated to lattice distortions in principle provides enough spatial diversity to separate the probe from the sample contribution, in principle.”

And later on, in the Method Section (3 – Inversion parameters):

“(alternatively, other illumination functions were used as an initial guesses and successful reconstructions could be produced, showing the robustness of the inversion process, see Supplementary information S1).”

Comments #3: “The authors developed the algorithm that allows to reconstruct the probe together with the sample structure. Approach is based on the use of kinematical equation (1) of the main text. Thus reconstructed probe has the size of $5 \times 6 \mu\text{m}^2$. If we estimate the speckle size of such a probe according to known equation $\lambda * L/D$, where λ is the wavelength, L is the sample-detector distance, and D is the area it will give us for a speckle size $43 \times 36 \mu\text{m}^2$ that is definitely less than the pixel size of detector that is for Maxipix is $55 \times 55 \mu\text{m}^2$. It means that probe is not sufficiently sampled to this size! The authors should explain how they go around this important point in their probe reconstruction.”

This comment shows that our description has to be improved to avoid confusion. The reconstructed probe size obtained in this manuscript is closer to $4 \times 5 \mu\text{m}^2$ (HxV), than $5 \times 6 \mu\text{m}^2$. In addition, attention must be paid that the Bragg geometry results in different treatments along the horizontal and vertical direction.

Along the horizontal direction, the probe size (*i.e.*, p_2 direction in Fig. 2a) is purely determined by the sampling of the detector pixel pitch size $u=55\mu\text{m}$ ($\lambda * L/u = 3.94 \mu\text{m}$). It was previously rounded to $4 \mu\text{m}$, but to be more precise, we changed it to **3.9 μm** in the revised version. Note that this value is big enough to cover a reasonably large region of the probe, including the central lobe and its tails, along the horizontal direction.

Along the vertical direction, the probe size is co-determined by the rotational angular step and the detector pixel pitch size due to the inclined geometry (see figure 2a). Given the Bragg angle for our measurement, the size is determined by the sampling of the effective rotational angular step $\Delta\theta=0.005^\circ/3$ after the angular upsampling by 3 times. To be more precise, we updated its value to **5.1 μm** (using the mathematic relation given in Eq. (2)). Here, the introduced up-sampling allows us to access to a bigger extent of the probe along the vertical direction, which greatly mitigates the aliasing effect caused by the experimentally designed angular under-sampling. This breakthrough was possible thanks to the use of the regularization applied along the propagation direction of the probe.

The revised manuscript now reads (Results section): ‘Note the extent of the field of views, of about $0.6 \times 6 \times 6 \mu\text{m}^3$ ($z \times y \times x$) for the 3D sample and **$5.1 \times 3.9 \mu\text{m}^2$ ($p_1 \times p_2$)** for the probe cross-section. **The probe field of view is therefore large enough to cover most of the probe extent, including the central lobe and the tails (as shown in Fig. 2b).**’.

Comments #4: “To my understanding ptychography scans were performed at different Bragg angular positions. For each angle sample was returned to its initial position and ptychography scans were repeated. For featureless sample as investigated in this work it is probably OK, but for more diverse sample it may be not a sufficient way of performing experiment, as step motors never come to the same position with the nanometer precision. Comment of the authors will be important to clarify this point.”

Indeed, Bragg ptychography has very high requirement on the repeatability of the scanning positions chosen under different incident angles, which is very challenging for real experiments due to possible mechanical instabilities. Here for this experiment, we made very good efforts to align and stabilize our setup prior to the Bragg ptychography data-set acquisition. Thereby, we managed to get a relevant dataset. This is demonstrated by the quality of the reconstruction, which is able to evidence localized defects such as dislocations.

To go further along these lines, let us note that this requirement can, in principle, be relaxed by the new back-projection reconstruction algorithm (see, e.g. [S. O. Hruszkewycz *et al.*, Nature Materials 16, 244 (2017)]), proposed recently by some of us. This formalism allows adopting position refinement strategies, both for the angular positions and for the beam-to-sample positions. This has been already demonstrated for angular uncertainties [I. Calvo_Almazan *et al.*, Scientific Reports 9, 6386 (2019)] and we are now addressing the case of the sample position (on-going project).

To address this comment, we have added the following statement in the Methods section:

“The Bragg ptychography acquisition was performed after ensuring the thermal stability of the experimental hutch was reached and assessing the mechanical stability and repeatability of the set-up.”

And later on, in the discussion part:

“Note that the observations of their sharp structure confirms the good quality of the data-set including possible beam-to-sample position uncertainties.”

Comments #5 and #6: *“The authors perform Bragg angular scans of their sample at different positions. I will be thankful if they will show the rocking curve in the region of pure Si wafer and in the region of the implanted part of the sample also indicating angles at which ptychography measurements were performed” and “Related to that, I will be thankful if the authors will show reciprocal space maps collected in their experiment also both in Si wafer and implanted parts of the sample.”*

We understand the rationale behind these two comments, which aim at completing the information regarding the Bragg ptychography data-set and its quality. While the Reviewer mentioned pure Si wafer, we believe he/she refers to the unimplanted region, which is a W crystalline grain (indeed, there is no pure Si wafer in the sample we investigated).

To answer the Reviewer’s comments regarding the rocking curves, we have added a new Supplementary figure (new Fig. S1), which presents two full rocking curves acquired prior to the Bragg ptychography measurement, respectively obtained in the non-implanted and implanted regions. To complete this plot, we have added the rocking curve extracted from the Bragg ptychography data-set, for two chosen positions, in the non-implanted and implanted regions.

To answer the Reviewer’s comments regarding the rocking curves, we have added in the same figure S1, two 3D plots of the intensity distribution extracted from the full Bragg ptychography data-set, at the same sample positions as the ones chosen for the rocking curves.

The text now reads (Supplementary information):

“**Fig. S1 - Experimental data-set.** (a) Complete rocking curves for both unimplanted and implanted regions, measured prior to the Bragg ptychography acquisition. The gray rectangle indicates the angular range chosen for the Bragg ptychography measurements. (b) The rocking curves at positions #381 and #400 extracted from the Bragg ptychography data-set. Note that the complete rocking curves were not taken at exactly the same sample positions as #381 and #400 from the Bragg ptychography measurements. (c) 3D rendering of the intensity distribution (iso-surface) obtained at position #381 during the Bragg ptychography acquisition and plotted in an orthogonal frame. One detector plane position is highlighted by a dashed rectangle. (d) 2D intensity distribution shown in the plane highlighted in (c). (e) and (f) same as (c) and (d), respectively, for position #400.”

Some minor points:

Comments #m1: “In the 3-rd paragraph where the authors describe different ptychography measurements performed for different samples they also should cite the work: Dzhigaev et al., ACS Nano, 11, 6605–6611 (2017), where ptychography approach was applied to imaging the strain field in Nanowires.”

Thanks for drawing our attention to this missing reference. We have added the suggested citation and modified the text as following:

‘... stacking faults and strain fields in semiconductor quantum wire [Hill, Dzhigaev] and...’

Comments #m2: Line 130. The authors should write that experiment was performed at ESRF ID01 beamline.

Revised accordingly to Referee’s suggestion. The text now reads:

“BP microscopy was performed at ESRF ID01 (a third generation synchrotron beamline)...”

Comments #m3: “On lines 131-132 the authors should provide correct focus size values at the sample (taking also into account Bragg angle).”

Thanks for these comments, which show that the values related the probe size characterization should be better documented. In order to provide a full description, we first included the probe size information, in the plane perpendicular to the beam. At the beginning of the Results section, the text now reads:

“An 8 keV coherent x-ray beam was focused down to about 400 x 200 nm (horizontal versus vertical FWHM of the central lobe in the plane perpendicular to the beam) at the sample position...”

To further take into account the Bragg angle, we think the better place to provide this information is in the ‘experimental details’ of the Methods section. Therefore, we added the following sentence:

“This inclined geometry results in an elongated footprint of the central beam spot with a size of 400×288 nm² (H×V).”

Comments #m4: *Lines 137 and 140. Words “sufficiently small” and “sufficiently large distance” should be substituted by the numbers.*

The is done accordingly to Reviewer’s suggestion, however still preserving the rationale behind the chosen values. The text now reads (in the Results section):

“in steps of 100 nm, sufficiently small to collect partially redundant information (corresponding to 75% and 65% overlapping along the two scanning directions, once the footprint elongation of the probe central lobe is taken into account for Bragg geometry, see Method section).”
“at a distance of 1.4 m from the sample, large enough to ensure...”.

Comments #m5 *“In Eq. (1) $\rho(r)$ is not specified after equation.”*

Thanks for pointing that mistake out. This is now corrected and reads:

“An effective complex-valued crystalline electron density, $\rho(\mathbf{r})$, specifically designed...”

Comments #m6: *“I think also that authors should be careful with using the term “electron density”, but use instead rather “crystalline electron density”. This is especially valid for the results presented for example in Figs. 3 and 4, where the bottom left part of the sample is “invisible” due to stacking fault. However electron density is not zero in this region as it is well seen in Fig. 1a.”*

This is a very good suggestion. We have changed all the “electron density” to “crystalline electron density”. This was done in seven places along the text.

Comments #m7: *“Fig. 1. In the inset of Fig. 1a scale bar should be given. In panel Fig. 1b box of monochromator unit should be substituted by two crystals. Coordinate system p_1, p_2, p_3 , should be explained in Fig. caption.”*

Thanks for pointing those weaknesses out. We corrected the manuscript accordingly:

- In Fig. 1a, we have added the scale bar.
- In Fig. 1b, we have changed the monochromator to two crystals.
- The probe coordinate system has been defined in the Fig.1 caption, as follows:
‘The sample frame (x, y, z) and the probe frame (p_1, p_2, p_3) are defined.’

Comments #m8: *“Fig. 2. Line 852. In Fig. caption, θ_1 is missing.”*

Thanks for this careful reading. “ θ_1 ” has been added in the caption.

Comments #m9: *“Fig. 3. It is not clear at which depth panels (c, d, f, g) are taken. It should be clearly indicated in Fig. caption. In panel(c, d) and (f, g) axes should be provided. In panels (d,g) the authors show the phase map from the reconstruction that has low significance due to wrapping of phase. It will be important if they will show the displacement field at least in some region of reconstructed part of the sample, then it will be easier to understand the flat strain fields shown in next Fig.”*

Figure 3 has been revised according to Reviewer’s suggestions.

- The depth for panels (c, d, f, g) is now given in the figure caption as follows:

“... shown over the plane indicated by red dashed rectangles at $z = 4.7 \mu\text{m}$.”

- The axes have been added to panel (c, d) and it should be implicitly clear for panel (f, g).

In addition, another figure has been added in the Supplementary Information. It is entitled ‘**Figure S5: 3D displacement extracted from the ϕ_{220} phase map of the object.**’ It contains the displacement field extracted from the unwrapped phase map, as suggested by the referee. In the main text, it is introduced in the Result section as:

“To this aim, we used 3D phase map (an unwrapped version of it being converted into the displacement field component and presented in Supplementary information S5) to extract the lattice strain projected...”

Reviewers' comments:

Reviewer #1 (Remarks to the Author):

I sincerely admire authors' great efforts for their response to the reviewers' comments. The manuscript has been improved for some points.

However, I still have the following concerns on the authors' claims. I think that the authors should make these point clearer.

(1) Issue on "invisible" defects

(a) The authors claim that "TEM is insensitive to defects smaller than ~ 1.5 nm. (l. 48)" However, I do not think that this is a fair statement, considering the authors' method to detect the "invisible" defects by the lattice strain. Even using TEM, the strain due to the "invisible" defects can be detected using various methods with spatial resolution being higher than authors' method. For example, the precision of CBED for the strain measurement reaches 10^{-4} , which is very similar to that in the authors' method.

(b) The authors claim that "These are too small to be imaged directly by TEM, as confirmed by TEM observations of tungsten implanted with He under similar conditions in [D.E.J. Armstrong et al., Appl. Phys. Lett. 251901, 1 (2013)] (Response to Comment #3 by Reviewer #1)" However, this referred work only examined the He-vacancy complexes using TEM. I think that the mobile "invisible" SIA or SIA clusters might be able to make mutual interaction and grow to "visible" clusters.

(2) Data quality

(a) The authors responded to Comment #4 by Reviewer #1. However, even after reading their response, to my eyes, the strain distribution does not look homogeneous but it looks lower at the region near the grain boundary (new Fig. 5d). For arguing the defect homogeneity and the absence of the denuded zones, the authors need to put the error bars to the data.

Reviewer #2 (Remarks to the Author):

The authors have carefully address all the comments from the reviewers. The manuscript can be accepted.

Reviewer #3 (Remarks to the Author):

I appreciate the work that was done by the authors to address the questions raised by all referees.

About the title of the manuscript. I would strongly suggest to the authors to remove word 'invisible' from the title and change it to:
"Revealing nano-scale lattice distortions in implanted material with 3D Bragg Ptychography".

About publication of the manuscript. I would agree with another referee that this manuscript has to be submitted to materials structure studies or nuclear technologies journal rather than be published in Nature Communications journal.

RESPONSE TO REVIEWER #1

We reply below to all his/her comments and questions. The corresponding modifications in the text are highlighted in blue.

General comments: *«I sincerely admire authors' great efforts for their response to the reviewers' comments. The manuscript has been improved for some points. However, I still have the following concerns on the authors' claims. I think that the authors should make these points clearer. »*

We thank Reviewer #1 for this positive comment, and we are grateful that he/she leaves us the possibility to clarify some remaining concerns. In this revised version, we believe we have addressed these in a convincing manner as follows below:

Comments #1a and 1b: *“Issue on “invisible” defects*

(a) The authors claim that “TEM is insensitive to defects smaller than ~1.5 nm. (l. 48)” However, I do not think that this is a fair statement, considering the authors' method to detect the “invisible” defects by the lattice strain. Even using TEM, the strain due to the “invisible” defects can be detected using various methods with spatial resolution being higher than authors' method. For example, the precision of CBED for the strain measurement reaches 10^{-4} , which is very similar to that in the authors' method.”

(b) The authors claim “These are too small to be imaged directly by TEM, as confirmed by TEM observations of tungsten implanted with He under similar conditions in [D.E.J. Armstrong et al., Appl. Phys. Lett. 251901, 1 (2013)] (Response to Comment #3 by Reviewer #1)” However, this referred work only examined the He-vacancy complexes using TEM. I think that the mobile “invisible” SIA or SIA clusters might be able to make mutual interaction and grow to “visible” clusters.”

1(a) - Regarding the comparison of our method with electron-based microscopy approaches (e.g., TEM, CBED, etc...), we agree that the sensitivity to the strain can be similar. However, none of these approaches is able to provide 3D information in a thin (~50 nm) or thick sample (~2 μm), as we have noted in the second paragraph of the Introduction section. In order to provide complete comparison, we refer to Tables 1 and 2 below for an overview of their respective performances and the comparison with Bragg ptychography achievements. Table 1 is reproduced from B  ch   *et al.*, *Ultramicroscopy* **131**, 10

(2013). The strong complementarity of the electron-based and our X-ray based approaches can be assessed.

The use of CBED, as proposed by Referee #1, would result in an averaging of the strain information along the sample thickness direction (400 nm). As proven with our work, the resulting signal would be affected by the spurious strains caused by FIB damage that are present at both sides of the sample slab. The use of our 3D Bragg ptychography approach, which can distinguish the strains arising from the bulk of the sample with respect to the strain induced close to the surface by the FIB, has led to an estimate of the concentration of the invisible defects, and allowed us to establish that they are randomly oriented. These findings are unmatched to the best of our knowledge.

In order to better emphasize this point, the following text has been added in the Introduction part:

“Although electron-diffraction based microscopy techniques can also indirectly measure these small defects via strain at very high spatial resolution, they are often limited to 2D information and/or small field of views.”

And in the Discussion part:

“This strain distribution underlines the importance of probing the 3D information, a strategy that is highly challenging with electron-diffraction based microscopy approaches. While electron microscopy is able to provide strain maps with sensitivity comparable to X-ray Bragg diffraction (about 10^{-4}), the 2D electron microscopy image only provides strain information averaged along the sample thickness direction. The 3D approach dramatically simplifies data interpretation ...”

1(b) - The observation of visible clusters, arising from interactions between “invisible” SIA or SIA clusters, is a very interesting point. Ab-initio calculations (Ref [55] in our manuscript) have shown that the SIAs remain bounded to the He-V complexes. So, at room temperature, the SIAs are not free to move. This means they cannot cluster. As a result, no clearly identifiable SIA clusters or rather interstitial loops will be seen in TEM.

To clarify this point, we have added the following in the Discussion part:

“Ab-initio calculations [55] have shown that the SIAs remain bound to the He-V complexes. This means that, at room temperature, the SIAs are not free to move and cannot cluster.”

[Redacted]

Table 1 (from A. Béch  et al., *Ultramicroscopy* 131 (2013) 10–23): This table summarizes the performance and constraints of several electron-beam microscopy approaches, to be compared to the proposed x-rays 3D Bragg ptychography methods (Table 2).

Note the typo on the last line, where ‘mm’ should read ‘nm’, instead.

	3D Bragg ptychography
Data type	3D map
Strain precision Strain accuracy	About 10^{-4} About 10^{-4} (after calibration)
Spatial resolution Field of view	Between 10-80 nm Unlimited
Acquisition easiness	+ (and ++ at 4 th generation sources)
Data treatment	From a few hours to a few days
Data volume	Depends on imaged volume
Sample thickness	From 50 nm to 2 μm

Table 2: This table summarizes the performance and constraints of 3D X-ray Bragg ptychography. Characteristics for which Bragg ptychography is superior (i.e., more flexible) to electron-based microscopy approaches are highlighted in green.

Comments #2: “(2) Data quality (a) The authors responded to Comment #4 by Reviewer #1. However, even after reading their response, to my eyes, the strain distribution does not look homogeneous but it looks lower at the region near the grain boundary (new Fig. 5d). For arguing the defect homogeneity and the absence of the denuded zones, the authors need to put the error bars to the data.”

We are extremely grateful to Reviewer #1, who pushed us further on the question of the denuded zone. In order to provide a detailed answer to this comment, we performed a fit of the strain profile perpendicular to the grain boundary, taking into account the spatial resolution effect. Surprisingly, it was not possible to fit the strain profile with the two considered models: (i) a fully denuded region or (ii) a fully strained region. Instead, we clearly needed to introduce (iii) a partially denuded region, corresponding to a strain of about $2.07 \times 10^{-4} \pm 0.13 \times 10^{-4}$ and a thickness of about 72 ± 8 nm. Here, the error bars are estimated from the spatial variations observed in the strain profile from one location to another. This new fit has been introduced in the Supplementary Figure S9 (replacing the former Fig. S9c). The two extreme models ((i) and (ii) described above) are also shown.

The text has been modified according to these new findings:

- In the Abstract: “A partially defect-denuded region is observed close to the grain boundary.”
- In the Result section, we removed the reference to the grain boundary: “Besides these, ϵ_{zz} presents a rather homogeneous behaviour in the whole implanted layer (Fig. 5c-f).”
- In the Discussion part: “... this is not exactly the case. From a detailed fit of the strain profile, shown in Supplementary Information S9, we see evidence of only a partially defect-denuded region, corresponding to a partial release of the strain down to about $2.07 \times 10^{-4} \pm 0.13 \times 10^{-4}$ over a thickness of about 72 ± 8 nm (here, the error bars are the standard deviations arising from the spatial variations observed in the strain profile at different locations). One possibility is that defect size affects the degree of denuding, whereby small defects, which experience smaller elastic driving forces to sinks, lead to partially denuded zones.”
- In the Conclusion: “Surprisingly, we only found a partially defect-denuded region in the vicinity of the crystal boundary.”
- In the new Fig. S9c: the fit of the strain profile at the grain boundary is now presented. The caption has been changed accordingly: “(c) Detailed analysis of the strain profile in the implanted region at the grain boundary. The orange dots represent the one-dimensional experimental strain profile averaged over 8 pixels. The three lines are fits of the data considering a spatial resolution of 39 nm and assuming (i) no denuded region, (ii) a fully defect-denuded region (57 nm large) and (iii) a partially defect-denuded region (with strain of 2.08×10^{-4} and width of 70 nm). Only the partially defect-denuded region model is able to account for the strain variation at the boundary.”

RESPONSE TO REVIEWER #2

General comment: « *The authors have carefully address all the comments from the reviewers. The manuscript can be accepted.* »

We thank Reviewer #2 for this positive and final comment.

RESPONSE TO REVIEWER #3

The corresponding modifications in the text are highlighted in orange.

General comment: « *I appreciate the work that was done by the authors to address the questions raised by all referees.* »

We thank Reviewer #3 for this positive comment. We agree that the comments raised during the previous revision round were very helpful to improve the readability of the manuscript and we thank all reviewers for their involvement.

Comments #1: « *about the title of the manuscript. I would strongly suggest to the authors to remove word 'invisible' from the title and change it to: "Revealing nano-scale lattice distortions in implanted material with 3D Bragg Ptychography".* »

The title has been modified accordingly.

Comments #2: « *about publication of the manuscript. I would agree with another referee that this manuscript has to be submitted to materials structure studies or nuclear technologies journal rather than be published in Nature Communications journal.* »

As a final comment, Reviewer #3 raises the question of the relevance of our manuscript to Nature Communications (a point he/she did not raise previously). Papers published by Nature Communications are expected to represent important advances of significance to specialists within each field (<https://www.nature.com/ncomms/about>). As such, we believe that Reviewer #3 in fact provided good reason to consider publication of our manuscript in Nature Communications in his/her previous report:

- « *This work reports on Bragg ptychography measurements of He implanted tungsten samples. The authors performed a complicated experiment and detailed analysis of the large amount of data collected in this experiment and report on strain fields observed in the tungsten sample in different regions of this sample: implanted and not implanted.* »

- « *... authors present a new way of reconstructing the probe of the illumination together with the reconstruction of the sample. That is a good achievement ...* »

In addition, with the advent of 4th generation synchrotron sources, we are expecting that Bragg ptychography will be greatly facilitated, as our recent results (not yet published, manuscript in preparation) has demonstrated: we have shown that 4th generation synchrotron sources allow one to perform Bragg ptychography experiments within a strongly decreased acquisition time (by a factor of about 100), as expected. However, and more importantly, the quality of the diffracted signal allowed us to perform high-quality Bragg ptychography images *on a non-optimized scanning stage* (including drifts, positioning errors and fly-scanning), a bottleneck, which has inhibited the application of Bragg ptychography, so far. This leap in robustness and the practical applications of Bragg ptychography are enabled by the advances described in this current manuscript, *i.e.*, the simultaneous retrieval of the probe and the angular-upsampling approach.

We anticipate that the impact of our current work will be strongly reflected in the years to come within the field of strain microscopy and coherent diffractive imaging. As Bragg ptychography is a key ingredient in the portfolio of many synchrotron beamlines worldwide, we anticipate that the quality of the results we obtained here on the complex implanted Tungsten polycrystalline sample, together with the availability of new sources and set-ups will strongly help to promote the development of Bragg ptychography on complex systems towards a broad readership. This is in tandem with the development of more sophisticated inversion algorithms like the one we demonstrated here.

REVIEWERS' COMMENTS

Reviewer #2 (Remarks to the Author):

The quality of the revised version has been improved. All the comments are carefully addressed. The manuscript is recommended to be published as it is.

Responses to the referees

Reviewer #2 (Remarks to the Author):

Comment #1: « The quality of the revised version has been improved. All the comments are carefully addressed. The manuscript is recommended to be published as it is. »

Answer #1: Thanks a lot.